# Condensin I$^{DC}$, DPY-21, and CEC-4 maintain X chromosome repression in *C. elegans*

**Jessica Trombley, Audry I. Rakozy, Christian A. McClear, Eshna Jash, Györgyi Csankovszki***

Department of Molecular, Cellular, and Developmental Biology, University of Michigan, Ann Arbor, Michigan, United States of America

* gyorgyi@umich.edu

## Abstract

Dosage compensation in *Caenorhabditis elegans* equalizes X-linked gene expression between XX hermaphrodites and XO males. The process depends on a condensin-containing dosage compensation complex (DCC), which binds the X chromosomes in hermaphrodites to repress gene expression by a factor of 2. Condensin I$^{DC}$ and an additional five DCC components must be present on the X during early embryogenesis in hermaphrodites to establish dosage compensation. However, whether the DCC's continued presence is required to maintain the repressed state once established is unknown. Beyond the role of condensin I$^{DC}$ in X chromosome compaction, additional mechanisms contribute to X-linked gene repression. DPY-21, a non-condensin I$^{DC}$ DCC component, is an H4K20me2/3 demethylase whose activity enriches the repressive histone mark, H4 lysine 20 monomethylation, on the X chromosomes. In addition, CEC-4, a protein that tethers H3K9me3-rich chromosomal regions to the nuclear lamina, also contributes to X-linked gene repression. To investigate the necessity of condensin I$^{DC}$ during the larval and adult stages of hermaphrodites, we used the auxin-inducible degradation system to deplete the condensin I$^{DC}$ subunit DPY-27. While DPY-27 depletion in the embryonic stages resulted in lethality, DPY-27 depleted larvae and adults survive. In these DPY-27 depleted strains, condensin I$^{DC}$ was no longer associated with the X chromosome, the X became decondensed, and the H4K20me1 mark was gradually lost, leading to X-linked gene derepression (about 1.4-fold). These results suggest that the stable maintenance of dosage compensation requires the continued presence of condensin I$^{DC}$. A loss-of-function mutation in *cec-4*, in addition to the depletion of DPY-27 or the genetic mutation of *dpy-21*, led to even more significant increases in X-linked gene expression (about 1.7-fold), suggesting that CEC-4 helps stabilize repression mediated by condensin I$^{DC}$ and H4K20me1.

## Author Summary

In some organisms, whether an individual becomes male, female, or hermaphrodite is determined by the number of their sex chromosomes. In the nematode *Caenorhabditis elegans*, males have one X chromosome, whereas hermaphrodites have two X chromosomes. This difference in the number of X chromosomes is crucial for deciding whether

**Data availability statement:** RNA seq data files are available from GEO database, accession number GSE262451. Scripts used to analyze data can be found at: https://github.com/eshnaj/trombley_X_repression_maintain_paper

**Funding:** This work was supported by the National Institute of General Medical Sciences (https://www.nigms.nih.gov/), grants number R35GM149543 to GC, and R01 GM13385801 to GC, and the National Science Foundation (https://www.nsf.gov/) grant number MCB 1923206 to GC. Some C. elegans strains used in this study were obtained from the Caenorhabditis Genetics Center (CGC), which is funded by the NIH Office of Research Infrastructure Program (P40 OD010440). The funders had no role in the study design, data collection and analysis, decision to publish, or preparation of the manuscript.

**Competing interests:** The authors have declared that no competing interests exist.

an individual becomes a hermaphrodite or a male. However, having two X chromosomes can lead to problems because it results in different gene expression levels, resulting in hermaphrodite lethality. To solve this issue, many organisms undergo a process called dosage compensation. Dosage compensation in *C. elegans* is achieved by a group of proteins known as the dosage compensation complex (DCC), which includes a protein called DPY-27. The function of DPY-27 is essential during early embryonic development. This study shows that in contrast to early embryonic development, larvae and adults can still survive when DPY-27 is missing. In these worms, all known mechanisms involved in dosage compensation are disrupted and the X is no longer repressed. Our results suggest that the maintenance of dosage compensation in nematodes is an active process, and that it is essential for survival when the organism is developing, but once fully developed, the process becomes dispensable.

## Introduction

Chromosome conformation, subnuclear localization, and the post-translational modifications of histones are closely linked to gene expression. The relationship between these processes plays a crucial role in an organism's development, growth, and survival. Dosage compensation in the nematode *Caenorhabditis elegans* offers an excellent paradigm for exploring chromosome-wide gene regulation.

There is an inherent imbalance in chromosomal gene expression in species with heterogametic sexes [1]. As in numerous other species with chromosome-based sex determination, *C. elegans* has evolved several molecular mechanisms to ensure equal gene expression from the sex chromosomes between the sexes [2]. This process is known as dosage compensation. In *C. elegans,* hermaphrodites have two X chromosomes (2X:2A), whereas males have only one X and no other sex chromosome (1X:2A) [3]. This difference in chromosome count is essential for sex determination [3,4]. However, without correction, in hermaphrodites, transcription from the X doubles relative to males, leaving the hermaphrodite population inviable [5]. The dosage compensation mechanism in *C. elegans* corrects for the differing numbers of sex chromosomes by reducing the gene expression of the hermaphrodite X chromosomes [6]. Dosage compensation requires orchestrating several gene-regulatory events across X chromosomes in hermaphrodites to decrease gene expression by half.

A core component of the dosage compensation machinery in *C. elegans* involves a specialized condensin complex. Condensins are highly conserved protein complexes used in most cellular life, from prokaryotes to metazoans [7]. Condensin complexes are characterized by their ring-shaped structure, and they hydrolyze ATP to extrude DNA strands through the ring, thus leading to DNA coiling and loop formation [8]. This loop extrusion activity is thought to be essential for chromosome compaction in mitosis and meiosis [9]. Multicellular eukaryotes have two condensin complexes, condensin I and II, functioning in mitosis and meiosis [10]. Similarly, *C. elegans* utilizes condensin I and II complexes for chromosome compaction and segregation [11]. However, *C. elegans* has a third condensin complex that specifically functions in dosage compensation of the X chromosomes of hermaphrodites. The hermaphrodite-specific condensin is known as condensin I$^{DC}$ (Dosage Compensation) [11,12]. Condensins are composed of two structural maintenance of chromosomes (SMC) proteins and three chromosome-associated polypeptides (CAP) [13,14]. Condensin I$^{DC}$ earns its name due to the four shared protein subunits with condensin I. These subunits include the structural maintenance of chromosomes (SMC) protein MIX-1 and the chromosome-associated

polypeptides (CAPs) CAPG-1, DPY-26, and DPY-28. The difference between condensin I$^{DC}$ and condensin I is the second SMC subunit, DPY-27 in condensin I$^{DC}$ and its paralog SMC-4 in condensin I [11,15,16]. Condensin I$^{DC}$ compacts the dosage compensated X chromosomes; thus, the X occupies less nuclear volume in hermaphrodites than in males [17,18]. At a higher resolution, condensin I$^{DC}$ remodels the architecture of the X chromosomes to form topologically associating domains (TADs) characterized by a higher frequency of *cis*-interactions within domains [19,20].

Beyond condensin I$^{DC}$, the complete dosage compensation complex (DCC) in *C. elegans* includes the proteins SDC-1, SDC-2, SDC-3, DPY-30, and DPY-21 [11,21–23]. Together, these ten proteins bind and spread across the X chromosomes in hermaphrodites, inhibiting RNA Pol II binding, thus lowering transcriptional output by half [24]. One of the non-condensin DCC subunits is DPY-21 [22]. It contains a Jumonji C domain found in histone demethylases. DPY-21 has been shown to demethylate dimethylated histone 4 lysine 20 (H4K20me2) [25], previously deposited on histones by the activities of SET-1 and SET-4 [26,27], to the monomethylated form (H4K20me1). Thus, by the X-specific demethylase activity of DPY-21, H4K20me1 is enriched on the X chromosomes in hermaphrodites and contributes to further repression [25–27]. In addition to its demethylase function, DPY-21 has an additional uncharacterized role in dosage compensation [25,28]. Like other DCC members, DPY-21 activity is required to regulate the X chromosome effectively, as evidenced by the elevated X-linked gene expression in hermaphrodites with *dpy-21* mutations [25,29]. However, DPY-21 is not necessary to recruit the other DCC members, and the loss of its function leads to mild defects in hermaphrodite *C. elegans* but no significant lethality [22].

The nuclear lamina protein CEC-4 and genes required for heterochromatin formation provide an additional layer of gene regulation in *C. elegans,* impacting dosage compensation. In *C. elegans,* heterochromatic regions of chromosomes are located near the nuclear periphery [29,30]. Histone methyl transferases (HMTs) MET-2 and SET-25 target H3 lysine 9 (H3K9) on chromatin to deposit mono-, di-, and trimethylation [30]. The lamina-associated protein CEC-4 binds to heterochromatic H3K9me regions on both the autosomes and the X [31,32]. The tethering of heterochromatic regions of the chromosomes to the nuclear lamina contributes to stabilizing cell fates during *C. elegans* development [32]. In addition, the loss of CEC-4 or heterochromatin leads to the decompaction and relocation of the X into a more interior position in the nucleus and a mild increase of X-linked gene expression without impacting autosomal gene expression, indicating dosage compensation defects [33]. Despite defects in the chromosomal organization, mutants with non-functional CEC-4 survive with no apparent phenotypes [32].

Dosage compensation is set up in a stepwise manner over the course of development in *C. elegans.* The process is initiated around the 40-cell stage, when the DCC subunit SDC-2 is expressed and localizes to hermaphrodite X chromosomes [34], which leads to the recruitment of the other DCC subunits [35]. This initiation step coincides with the loss of pluripotency of embryonic blastomeres [36,37]. DPY-21 localizes to the X after the initial loading of condensin I$^{DC}$, and H4K20me1 becomes enriched on the X between the bean and comma stages of embryogenesis [25,36]. By the end of embryogenesis, most cells exit mitosis [38], after which the dosage-compensated state is maintained in postmitotic cells. We refer to this stage as the maintenance phase. Earlier studies using cold-sensitive *dpy-27* alleles suggested that DPY-27 activity is essential during a critical developmental time window in mid-embryogenesis, after which loss of its function has a lesser impact on viability [39]. These results suggest that while condensin I$^{DC}$ is required to initiate dosage compensation, other mechanisms may be responsible for maintaining repression once established, including ones mediated by DPY-21 and CEC-4.

The requirement for CEC-4 also changes with the differentiation state of embryonic cells. CEC-4 is required to tether heterochromatic transgenic arrays to the nuclear lamina in embryos but not in L1 larvae [32]. At the larval stages and beyond, additional mechanisms, including ones mediated by MRG-1 and CBP-1, contribute to sequestering heterochromatin to the nuclear lamina [40]. However, X chromosome compaction and peripheral localization are affected in *cec-4* adults, even in fully differentiated postmitotic cells [33]. We hypothesize that CEC-4 stabilizes the repression of the X chromosomes during the maintenance phase of dosage compensation.

In this study, we used the auxin-inducible degradation (AID) system to deplete the condensin I$^{DC}$ subunit DPY-27 at various stages of development. We assessed the importance of condensin I$^{DC}$ to X chromosome repression and organismal viability during the establishment and maintenance phases of dosage compensation. Our findings reveal that the continued presence of DPY-27 is required to maintain X chromosome repression after initial establishment. However, continued repression is less essential for viability during the maintenance phase. Additionally, we evaluated the hypothesis that the nuclear lamina protein CEC-4 helps stabilize the repressed state by combining *cec-4* mutations with depletion of DPY-27 or mutations in the histone demethylase *dpy-21*. We show that the loss of CEC-4 exacerbates the defects caused by the lack of DPY-27 or DPY-21 function. Our results reveal the differential contributions of the various repressive pathways to X chromosome dosage compensation.

## Results

### The impact of DPY-27 depletion on hermaphrodite embryos and larvae

Using a strain containing an auxin-inducible degron (AID)-tagged *dpy-27* allele [41] and expressing the plant F-box protein TIR1 under a ubiquitous promoter (*eft-3*) [42] (referred to as *TIR1; dpy-27::AID*, for complete genotype and allele information for all strains see the Materials and Methods), we were able to time the depletion of DPY-27 by moving worms to auxin-containing plates during or after embryogenesis. To evaluate whether the lack of CEC-4 exacerbates the defects, we also generated a strain containing the *cec-4 (ok3124)* mutation in addition to *TIR1* and *dpy-27::AID* (referred to as *TIR1; dpy-27::AID; cec-4)*. For controls, we used wild-type worms (N2), *cec-4* mutants, and a strain with AID-tagged *dpy-27* but no TIR1 expressing transgene (referred to as *dpy-27::AID)*.

Our first experiment investigated the ability of embryos exposed to auxin to survive in the absence of DPY-27 throughout the entirety of embryonic development. To ensure that the maternally loaded DPY-27 is also depleted, we initiated our assay at the L4 larval stage of the parent before oocyte production begins (Fig 1A) [15]. L4 larvae were placed on auxin-containing plates for a 24-hour egg-laying period, after which the parents were removed. This procedure ensures that both *in-utero* and *ex-utero* development occurs in the presence of auxin. Strains not exposed to auxin were mostly viable, although the *dpy-27::AID*-containing strains had a very low, but statistically significant, level of lethality (Fig 1B, Fisher's exact test, p values listed in S1 File). While auxin treatment did not significantly impact the survival of control embryos (N2, *cec-4, dpy-27::AID)*, 100% of *TIR1; dpy-27::AID* and *TIR1; dpy-27::AID; cec-4* worms died as embryos or L1 larvae (Fig 1B). These results indicate that DPY-27 function during embryonic development is essential for the survival of hermaphrodite worms, which is consistent with previous studies [39].

At the onset of dosage compensation during early embryogenesis, condensin I$^{DC}$ loads onto the X chromosomes in hermaphrodites [15]. However, it is unclear whether the sustained presence of the complex is essential once dosage compensation is established. To assess the requirement for DPY-27 to remain on the X during larval development, we exposed our strains to auxin starting at the L1 larval stage for a period of three days. We compared survival

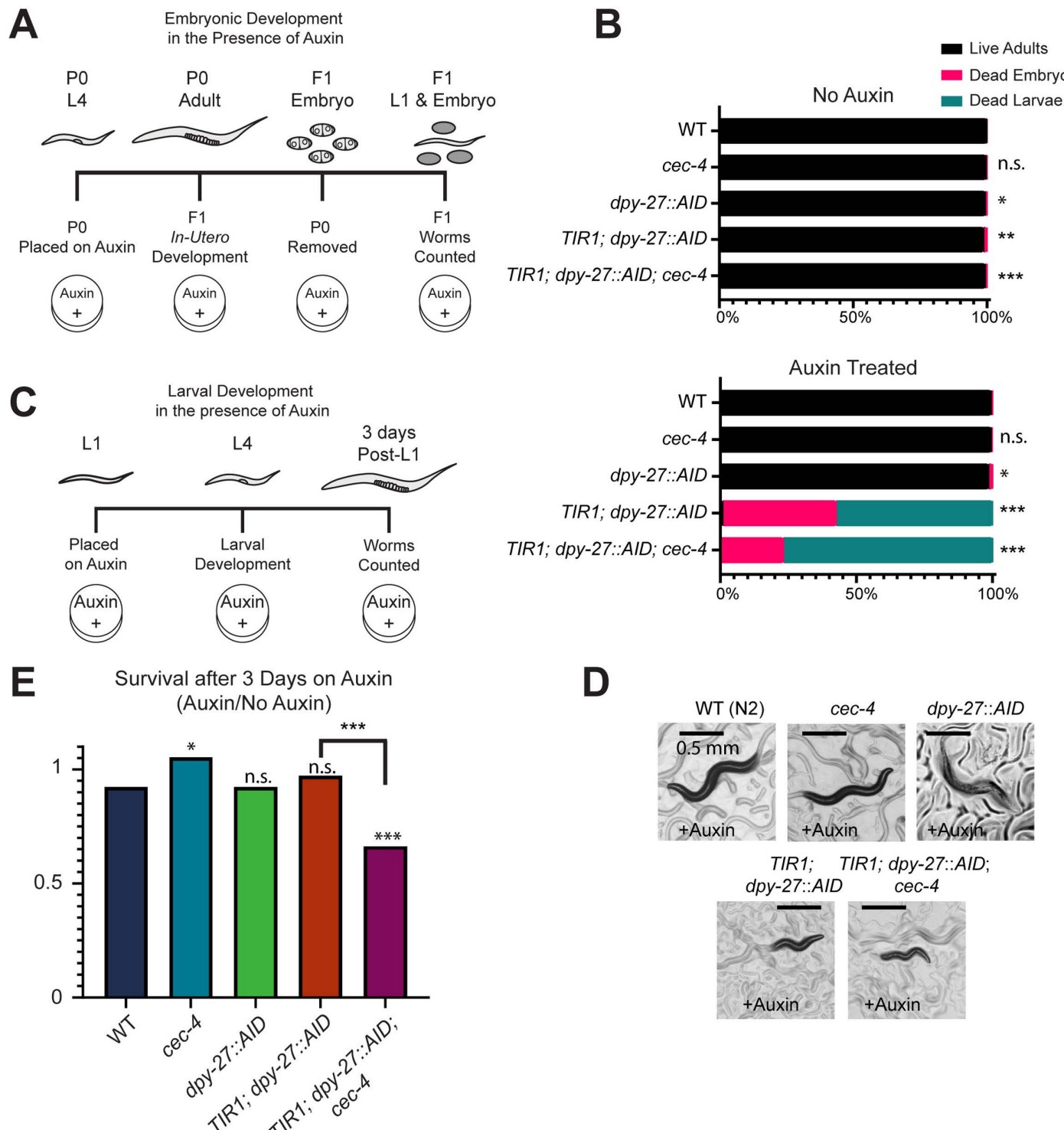

**Fig 1. DPY-27 depletion leads to lethality and developmental defects.** (A) Overview of the timeline of auxin exposure initiated from parental age L4 through F1 development. (B) Auxin exposure throughout embryogenesis results in compromised viability in *TIR1; dpy-27::AID* and *TIR1; dpy-27::AID; cec-4* strains. Total counts from three independent replicates are shown. N>600.Significance was determined by Fisher's exact test relative to wild-type values. Dead embryos and dead larvae were summed as one value for statistical analyses. Complete statistical analysis can be found in S1 File. (C) Overview of the timeline of auxin exposure from the L1 stage for three days. N2 worms in this time frame grow to young adults. (D) Phenotypic images of hermaphrodites exposed to auxin from the L1 stage for three days. Scale bar, 0.5 mm. (E) Larval survival on auxin relative to survival off auxin. Total counts from three independent experiments are shown. N>600. Statistical significance was determined by Chi square tests. Complete statistical analyses can be found in S2 File. n.s. = not significant, * = p < 0.05, ** = p<0.01, *** = p < 0.001.

on auxin-containing plates to survival on plates without auxin (Fig 1C). Strains developing without auxin exposure grew to adulthood and produced viable offspring in the timeframe of the experiment. In contrast to embryonic exposure, exposing larvae to auxin had a much lesser effect on survival. Most *TIR1; dpy-27::AID* larvae survived similarly to N2, *cec-4*, or *dpy-27::AID* controls (Fig 1E, Chi-square test, p values listed in S2 File). However, DPY-27 depleted strains developed slowly and exhibited developmental defects, including stunted growth (Fig 1D). The rare embryo produced by these hermaphrodites did not hatch. Adding *cec-4* to the background lowered the survival rate, suggesting that lack of both DPY-27 and CEC-4 results in more severe defects than lack of DPY-27 alone (Fig 1E).

## Exposure to auxin rapidly depletes DPY-27

We next investigated the degree of DPY-27 depletion in auxin-treated strains relative to untreated. DPY-27 was visualized in each strain by immunofluorescence (IF) after worms were exposed to auxin for three days starting at the L1 larval stage. Due to the underdeveloped nature of the DPY-27 depleted strains, we based the timing of IF staining on the growth of wild type. After 3 days, wild-type N2 worms have developed into young adults. IF images were captured in intestinal nuclei. Intestinal cells are 32-ploid and their large size facilitates analyses of the nucleus. Before auxin treatment, all our strains had a wild-type pattern of DPY-27 localization within the nucleus of intestinal cells (Fig 2A), consistent with the lack of any visible defects or lethality in these strains. Auxin treatment in N2, *cec-4*, and *dpy-27::AID* strains led to no observable changes. However, in the *TIR1*-containing strains, *TIR1; dpy-27::AID* and *TIR1; dpy-27::AID; cec-4,* auxin treatment resulted in depletion of DPY-27 to background levels (Fig 2A), suggesting that auxin-induced degradation was successful.

To quantify the level of depletion and to test reproducibility, we performed three independent experiments and imaged nuclei from each using standardized exposure times to allow direct comparisons between samples (Fig 2B and 2C). We included WT males in the analysis. WT males do not express DPY-27 [15], and the level of DPY-27 fluorescence measured in males was interpreted as background. Anti-HTZ-1 (a histone H2A variant) staining was used as control (S1 Fig). Variability between replicates is shown on S2 and S3 Figs. Quantification of nuclear DPY-27 fluorescence intensity over background fluorescence intensity showed that *TIR1; dpy-27::AID* and *TIR1; dpy-27::AID; cec-4* before auxin treatment had somewhat lower DPY-27 staining compared to WT, but significantly more than in males (**Fig 2C**). The somewhat reduced DPY-27 staining is consistent with a low level of lethality in these strains, and indicate that dosage compensation maybe slightly reduced compared to WT. In *TIR1; dpy-27::AID* and *TIR1; dpy-27::AID; cec-4* worms treated with auxin from L1 stage for three days, DPY-27 staining was indistinguishable from staining in WT males.

In the L1 to adult timeframe, intestinal nuclei undergo 4 rounds of endoreduplication and reach the stage of 32-ploidy [43]. To analyze the continued need for DPY-27 after these altered cell cycles have taken place and the cells have fully differentiated, we performed experiments where the auxin treatment began in adulthood (3 days past L1). These experiments also allowed to test the rapidity of auxin induced depletion of DPY-27. In previous studies in *C. elegans,* protein degradation was observable within 30 minutes [42]. We subjected adult hermaphrodites to auxin for one hour and for 24 hours. In *TIR1; dpy-27::AID* and *TIR1; dpy-27::AID; cec-4* strains, both treatments led to DPY-27 levels indistinguishable from what is seen in males (Fig 2B and 2C), suggesting that one hour is sufficient to deplete DPY-27 protein levels to background. Independent repetitions of these experiments showed that rapid depletion of the protein is reproducible (S2 and S3 Figs).

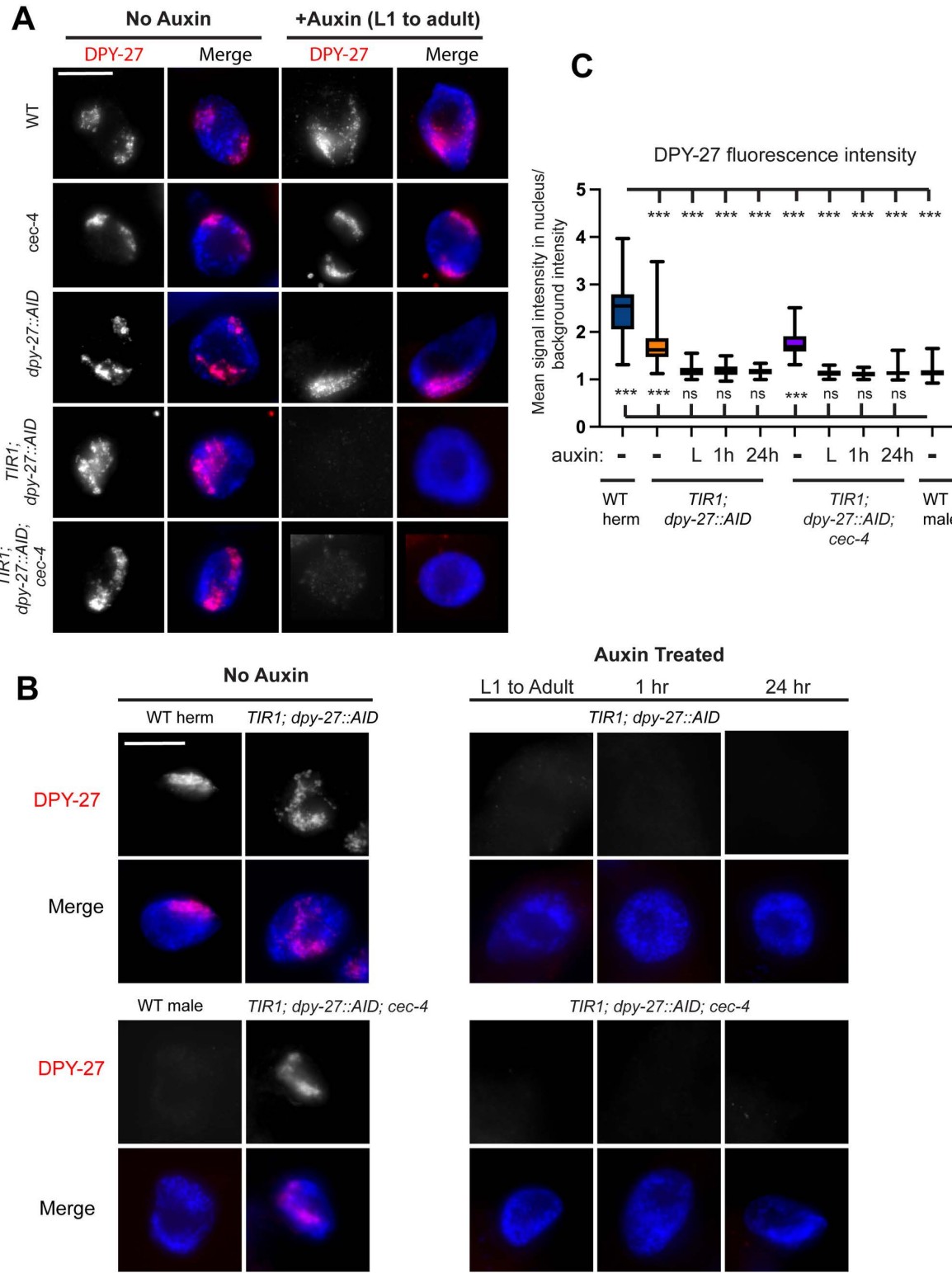

**Fig 2. Auxin treatment leads to depletion of DPY-27.** (A) Immunofluorescence staining of DPY-27 within the intestinal nuclei of young adult hermaphrodites. Maximum intensity projections of DAPI staining of DNA (blue) and DPY-27 signal (red) are shown. DPY-27 is present within the nuclei in strains not exposed to auxin, but exposure to auxin for 3 days starting at L1 results in a significant depletion of DPY-27 signal in strains that contain *TIR1* and *dpy-27::AID*. Original unaltered images are shown on S1 Fig. (B) Maximum intensity projections of nuclei stained with anti-DPY-27 antibodies after auxin exposure for various lengths of times (L1 to adult, 1 hour,

or 24 hours). Images were acquired with identical exposure times. Original unaltered images, with HTZ-1 costain, are shown on S1 Fig. Scale bars, 10 μm. (C) Quantification of DPY-27 fluorescence signal in experiments shown in (B). Quantification is based on measuring mean DPY-27 fluorescence in the nucleus, normalized to background level fluorescence, from between 38 and 50 nuclei total collected in three independent experiments. Worms were not exposed to auxin (-), exposed to auxin from L1 stage for three days (L), or exposed to auxin from day 1 of adulthood for 1h or 24 hours. Whiskers indicate min-max values. Statistical analysis was done using Brown-Forsythe and Welch ANOVA tests with multiple comparisons. Comparisons to WT hermaphrodites are shown above and comparisons to WT males are shown below the box plots. For p values and all pairwise comparisons see S3 File. n.s. = not significant, * = p < 0.05, ** = p<0.01, *** = p < 0.001.

## DPY-27 depletion disrupts DCC binding and stability

To assess the binding of other condensin I$^{DC}$ subunits to the X chromosome following DPY-27 depletion, we performed IF staining for the subunit CAPG-1 [11]. In WT and *cec-4* strains, CAPG-1 maintains a signal pattern indicative of X chromosome binding both with and without auxin treatment (Fig 3A). However, in *TIR1; dpy-27::AID* and *TIR1; dpy-27::AID; cec-4* after auxin treatment from the L1 stage for three days, the CAPG-1 signal becomes non-specific and diffuse throughout the entire nucleus (Fig 3A). Our results indicate that after prolonged auxin exposure, condensin I$^{DC}$ no longer maintains localization to the X chromosomes following DPY-27 depletion.

To determine how fast CAPG-1 dissociates from the X and to determine whether the protein diffuses throughout the nucleus or becomes unstable, we again performed auxin treatment for 1 hour and 24 hours on adult worms and used standardized exposure times to image nuclei to be able to compare signal intensities across samples (Fig 3B). We first quantified the enrichment of CAPG-1 in a region of the nucleus, which we interpret as the X chromosomes, where CAPG-1 is known to localize to in WT worms [11]. We used line intensity scan quantification to determine the difference between CAPG-1 signal intensity in the enriched region over non-enriched region (Fig 3C, see methods). CAPG-1 enrichment again was somewhat lower in *TIR1; dpy-27::AID* and *TIR1; dpy-27::AID; cec-4* strains prior to auxin treatment, similar to the slightly reduced DPY-27 staining in these strains. After 1 hour or 24 hours of auxin treatment, enrichment becomes indistinguishable from (or lower than) what is seen in males. Since the level of enrichment seen in males we interpreted as being due to background staining by the antibody, we do not believe that the statistically significantly lower values in the *TIR1; dpy-27::AID* and *TIR1; dpy-27::AID; cec-4* strains has biological significance. These results suggests that within 1 hour of the initiation of DPY-27 depletion, CAPG-1 is no longer noticeably enriched on the X chromosomes.

We also quantified total CAPG-1 fluorescence in these nuclei. Nuclei from non-auxin treated *TIR1; dpy-27::AID* and *TIR1; dpy-27::AID; cec-4* worms had CAPG-1 staining similar to WT hermaphrodites. Signal intensity was only partially reduced (*TIR1; dpy-27::AID*) or similar to WT hermaphrodites (*TIR1; dpy-27::AID; cec-4*) after 1 hour of auxin treatment. However, after 24 hours of auxin treatment in both strains, the signal became indistinguishable from the signal seen in males that lack condensin I$^{DC}$. Together with the enrichment analysis, these results suggests that the CAPG-1 protein persists somewhat longer than DPY-27 after auxin treatment, but the protein present is unable to localize to the X chromosomes and eventually degrades.

## H4K20me1 does not persist on the X chromosomes after DPY-27 depletion

We next assessed the enrichment of the histone modification H4K20me1 on the X. In wild-type N2 hermaphrodites, this histone modification is specifically enriched on dosage-compensated X chromosomes [26,27], and this pattern was maintained in *cec-4* and WT

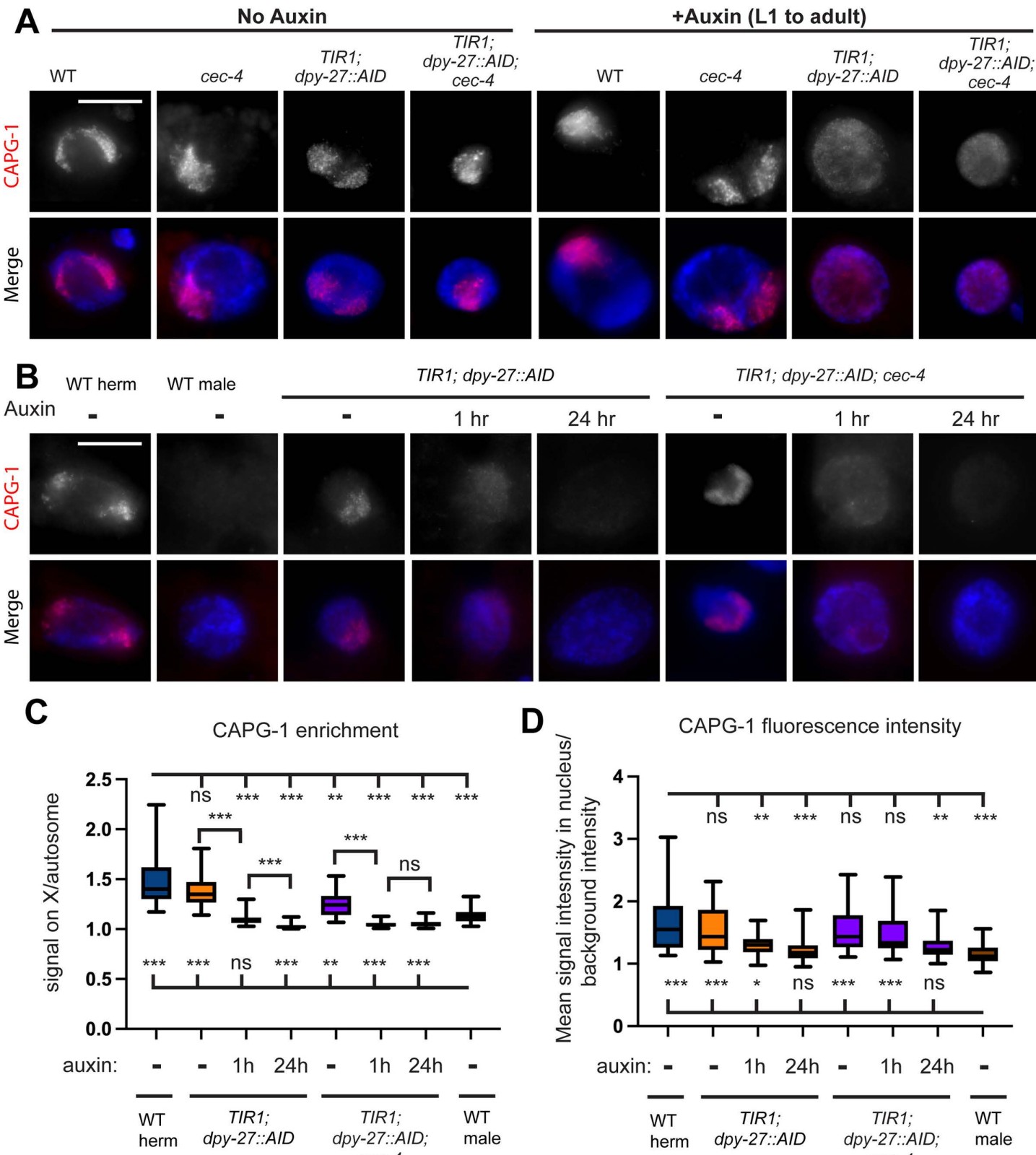

**Fig 3. In the absence of DPY-27, condensin I$^{DC}$ subunits become unstable.** (A) Maximum intensity projections of CAPG-1-stained intestinal nuclei of young adult hermaphrodites. DAPI staining of DNA (blue) merged with CAPG-1 signal (red) is also shown. DPY-27 is present within the nuclei in strains not exposed to auxin, but exposure to auxin for 3 days starting at L1 leads to lack of X enrichment of CAPG-1 signal in strains that contain *TIR1* and *dpy-27::AID*. Original unaltered images are shown on S4 Fig. (B) CAPG-1 staining after exposing young adults to auxin for 1 hour or 24 hours. Images were acquired with identical exposure times. Maximum

intensity projections are shown. Original unaltered images (with HTZ-1 staining control) are shown on S4 Fig. Scale bars, 10 μm. (C) Quantification of CAPG-1 fluorescence signal in the enriched region over non-enriched regions of the nucleus. Quantification is based on line-intensity scan analysis of 30-33 nuclei total collected in three independent experiments. Whiskers indicate min-max values. Statistical analysis was done using Brown-Forsythe and Welch ANOVA tests with multiple comparisons. Comparisons to WT hermaphrodites are shown above and comparisons to WT males are shown below the box plots. For p values and all pairwise comparisons see S4 File. (D) Quantification of mean CAPG-1 fluorescence intensity in the nucleus, normalized to background level fluorescence, from the same nuclei as in (C). Statistical analysis was done using Brown-Forsythe and Welch ANOVA tests with multiple comparisons. Comparisons to WT hermaphrodites are shown above and comparisons to WT males are shown below the box plots. For p values and all pairwise comparisons see S4 File. n.s. = not significant, * = p < 0.05, ** = p<0.01, *** = p < 0.001.

strains after auxin treatment. However, after exposure to auxin from the L1 stage for three days in *TIR1; dpy-27::AID* and *TIR1; dpy-27::AID; cec-4* strains, H4K20me1 was no longer specifically enriched in a region within the nucleus, resembling the pattern observed for CAPG-1 localization (Fig 4A). H4K20me1 first becomes enriched on dosage-compensated X chromosomes in mid-embryogenesis [27,36], well before the L1 stage, when auxin treatment was initiated. We conclude that the continued presence of DPY-27 and the DCC is required to maintain the enrichment of this chromatin mark on the X chromosomes during larval development.

To assess how fast H4K20me1 enrichment is lost in postmitotic nuclei after DPY-27 depletion, we treated adult worms with auxin for 1 hour and 24 hours and quantified enrichment and total fluorescence intensity of the H4K20me1 signal (Fig 4B). After 1 hour of auxin treatment, the signal remains clearly enriched in a portion of the nucleus in both *TIR1; dpy-27::AID* and *TIR1; dpy-27::AID; cec-4* strains. However, after 24 hours of auxin treatment, enrichment in both strains becomes similar to what is seen in males (Fig 4C). Assessment of total H4K20me1 signal intensity (Fig 4D) was less conclusive due to large cell-to-cell variation. Most genotypes had total fluorescence values not statistically different from WT hermaphrodites. Overall, we can conclude that enrichment of the H4K20me1 mark persists longer than the presence of condensin I$^{DC}$ on the X, but that enrichment does diminish within 24 hours. The *TIR1; dpy-27::AID* and *TIR1; dpy-27::AID; cec-4* strains appeared similar in this assay (S5 File), suggesting that the presence or absence of CEC-4 does not contribute to the maintenance of the H4K20me1 mark, at least at the resolution of immunofluorescence microscopy.

## The continued presence of DPY-27 is required to maintain X chromosome condensation

The X chromosomes of wild-type hermaphrodites occupy a lower proportion of nuclear space compared to what is predicted based on DNA content [17]. We previously showed that in DCC mutants and in *cec-4* mutant hermaphrodites, the X chromosomes decondense and that a larger proportion of the X signal is found in the central region of the nucleus [33]. The DCC starts to compact the X chromosomes in early embryogenesis [17]. We tested if the continued presence of DPY-27 is necessary for maintaining the compact conformation and peripheral localization of the X. We performed whole chromosome X paint fluorescence *in situ* hybridization (FISH) to mark the X chromosome territories in intestinal nuclei. Without auxin treatment, the X chromosome signal appeared as expected: compact and near the nuclear periphery in all strains without a *cec-4* mutation and less compact in strains with a *cec-4* mutation (Fig 5A). After auxin treatment from L1 stage for 3 days, the X chromosomes appeared enlarged in *TIR1; dpy-27::AID* as well. We quantified these changes by measuring the proportion of nuclear volume occupied by the X chromosome (see Methods). The X chromosomes occupied about 12-13% of the nuclear volume on average in wild type and close to 20% in *cec-4* mutants (Fig 5B) which is consistent with previous results [33]. In the *TIR1; dpy-27::AID* strain, depletion of DPY-27 led to decondensation of the X chromosome to a degree similar

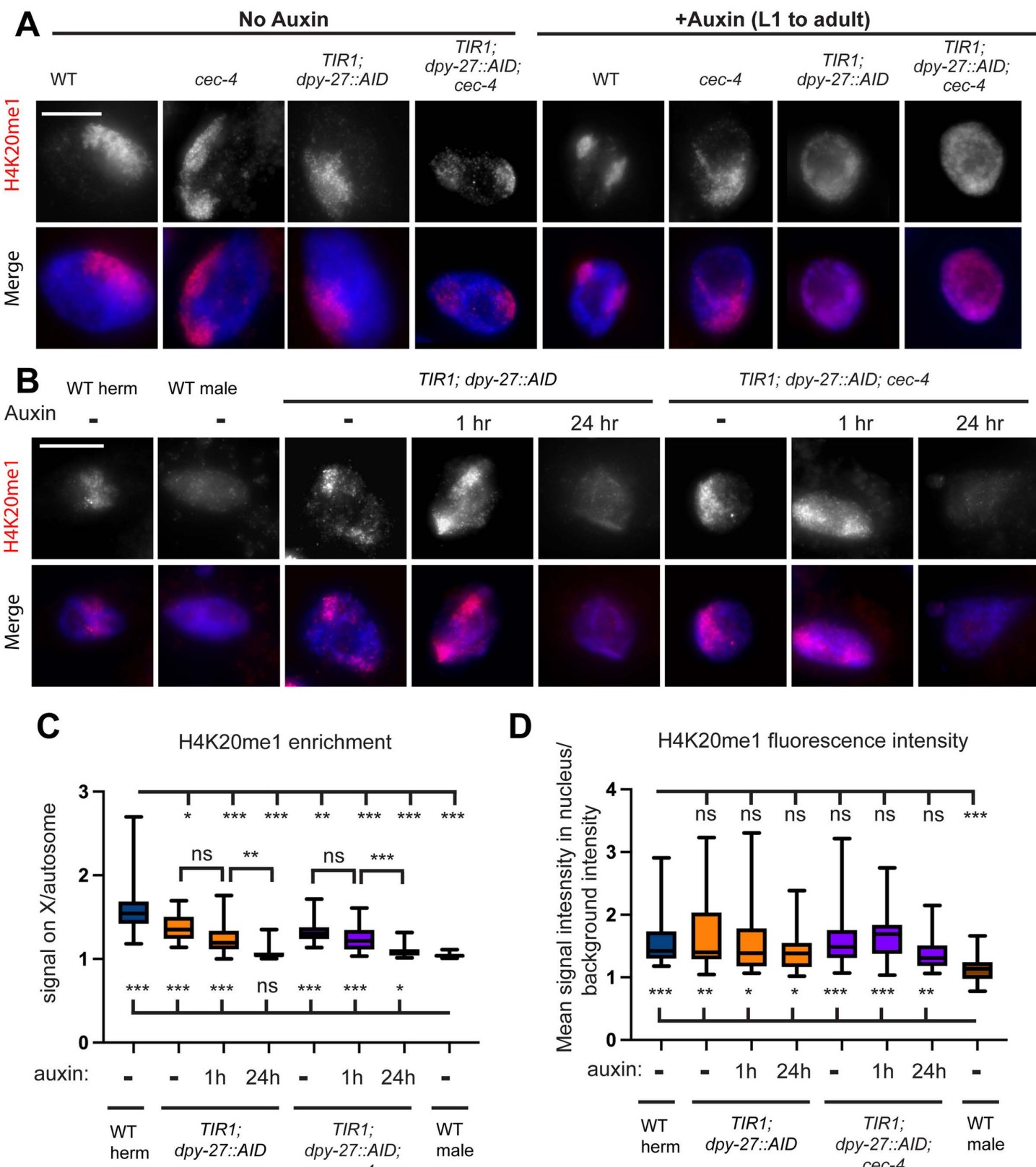

**Fig 4. In the absence of DPY-27, H4K20me1 enrichment is lost.** (A) Immunofluorescence staining of H4K20me1 within the intestinal nuclei of young adult hermaphrodites. Maximum intensity projections of DAPI staining of DNA (blue) and H4K20me1 signal (red) are shown. H4K20me1 is clearly enriched in a region within nuclei (assumed to be the X) in strains not exposed to auxin. Auxin exposure for 3 days starting at L1 led to lack of X enrichment of H4K20me1 signal in strains that

contain *TIR1* and *dpy-27::AID*. Original unaltered images are shown on S5 Fig. (B) Maximum intensity projections of H4K20me1-stained nuclei after auxin exposure for 1hr or 24 hours. Images were acquired with identical exposure times. Original unaltered images (with HTZ-1 staining control) are shown on S5 Fig. Scale bars, 10 µm. (C) Quantification of H4K20me1 fluorescence signal in the enriched region over non-enriched regions. Quantification is based on line-intensity scan analysis of 29-30 nuclei total collected in three independent experiments. Whiskers indicate min-max values. Statistical analysis was done using Brown-Forsythe and Welch ANOVA tests with multiple comparisons. Comparisons to WT hermaphrodites are shown above and comparisons to WT males are shown below the box plots. For p values and all pairwise comparisons see S5 File. (D) Quantification of mean H4K20me1 fluorescence intensity in the nucleus, normalized to background level fluorescence, from the same nuclei as in (C). Statistical analysis was done using Brown-Forsythe and Welch ANOVA tests with multiple comparisons. Comparisons to WT hermaphrodites are shown above and comparisons to WT males are shown below the box plots. For p values and all pairwise comparisons see S5 File. n.s. = not significant, * = p < 0.05, ** = p<0.01, *** = p < 0.001.

to *cec-4* mutants or to what was previously observed in worms treated with *dpy-27* RNAi [17]. These findings demonstrate the critical role of DPY-27 in maintaining the compact conformation of the X chromosome in wild-type hermaphrodites. In the *TIR1; dpy-27::AID; cec-4* strain, the X was already decondensed without auxin treatment due to the presence of the *cec-4* mutation, and depletion of DPY-27 led to a measurable level of additional decondensation (Fig 5B). These results indicate that the presence of DPY-27 maintains X compaction, and the additional loss off *cec-4* exacerbates the decondensation defects. To assess if shorter exposure to auxin in postmitotic cells of adults is sufficient to lead to X decondensation, we subjected young adult worms to 24 hours of auxin treatment. In both the *TIR1; dpy-27::AID* and the *TIR1; dpy-27::AID; cec-4* strains, this treatment led to similar levels of decondensation as seen after the longer L1 to adult exposure (**Fig 5B**). These results indicate that the continued the presence of DPY-27 is required to maintain a compact X structure even in fully differentiated postmitotic cells.

Compact dosage compensated X chromosomes are usually located near the nuclear periphery [33]. To assess any changes in subnuclear localization, we analyzed our X chromosome FISH signals in the various strains using a three-zone assay (see Methods) (Fig 5C)**.** The localization of the X within the nucleus was assessed in three concentric zones from a single focal plane taken from the middle of the nucleus, and the portion of the X signal in the central zone was quantified. In wild-type hermaphrodites, we observed the expected X localization. Only a small portion of the signal is found in the central zone, while the bulk of the X is located in the intermediate and peripheral zones of the nucleus (Figs 5C and S7) as observed before [33]. In *cec-4* mutants (*cec-4* and *TIR1; dpy-27::AID; cec-4)*, significantly more of the X signal is in the center, as expected. This central localization is even more significant in DPY-27 depleted strains (Fig 5C). In both *TIR1; dpy-27::AID* and *TIR1; dpy-27::AID; cec-4* strains, over 40% of the signal is localized in the central zone, suggesting that the depletion of DPY-27 has a more significant impact than the *cec-4* mutation. Adding a *cec-4* mutation did not significantly change X localization compared to DPY-27 depletion alone (S7 File). A corresponding shift away from the periphery was also observed, and a smaller proportion of the X signal was in the peripheral zone, although the differences were only statistically significant in *TIR1; dpy-27::AID* and *TIR1; dpy-27::AID; cec-4* strains after auxin treatment (S7B Fig and S7 File). Overall, these data indicate that the continued presence of DPY-27 is required to maintain the peripheral localization of dosage compensated X chromosomes. We note that some of the apparent relocation toward center could be a secondary consequence of decondensation, which allows the chromosome to occupy more internal regions.

## DPY-27 is needed to maintain repression of X-linked genes

We next tested whether these changes in condensin I^DC localization, X chromosome compaction, nuclear organization, and the depletion of repressive histone marks, led to increases in X chromosome gene expression in the *TIR1; dpy-27::AID* and *TIR1; dpy-27::AID; cec-4*

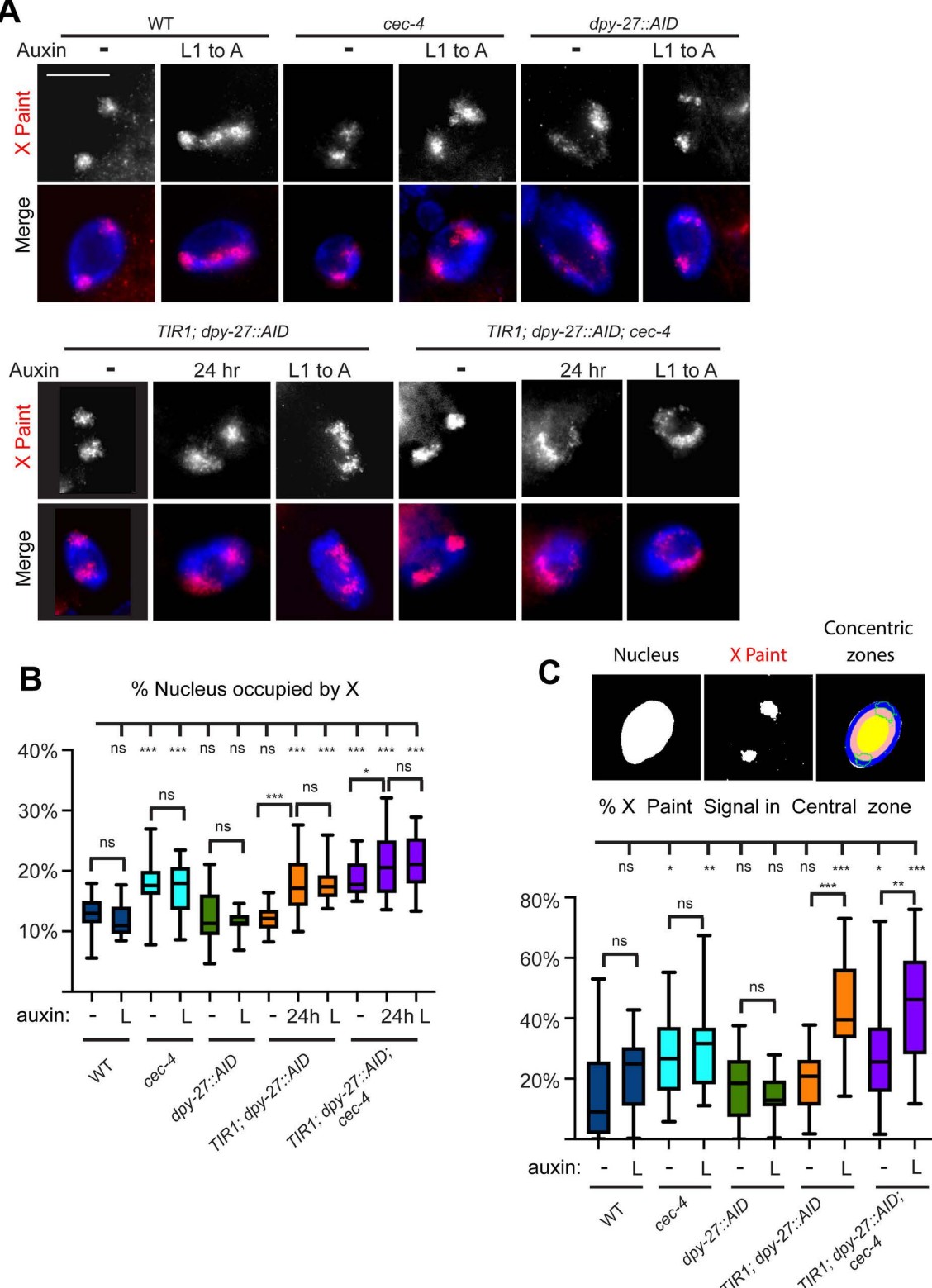

**Fig 5. The continued presence of DPY-27 is required for X compaction and peripheral localization within the nucleus.** (A) Intestinal nuclei from adult hermaphrodites were stained with whole X chromosome paint FISH probe (top row, red), and DNA was labeled with DAPI (blue). Images are of young adults not exposed to auxin, worms treated with auxin from L1 to adulthood, or young adults treated with auxin for 24 hours. Scale bars, 10 μm. Original unmodified images are shown on S6 Fig. (B) Quantifications

of the X chromosome volume normalized to nuclear size of worms grown auxin (-), worms treated with auxin from L1 stage for three days (L), and young adult worms treated with auxin for 24 hours (24h). Maximum intensity projections are shown in (A) and (B). (C) (top) Three-zone assay segmentation shown on a single slice from the middle of an intestinal nucleus. The amount of X signal (outlined in green) in each zone was quantified. (bottom) The proportion of the X paint signal seen in the central zone of the nucleus. In (C) and (D) 20-24 nuclei total from 3-4 independent experiments were quantified. Differences between samples were evaluated using unpaired Student's t-test. For complete statistical analysis see S6 and S7 Files. Analyses in all concentric rings can be found in S7 Fig. n.s. = not significant, * = p < 0.05, ** = p<0.01, *** = p < 0.001.

strains when treated with auxin. Although nuclear organization and H4K20me1 enrichment were not significantly different with and without a mutation in *cec-4*, the reduced viability of auxin-treated *TIR1; dpy-27::AID; cec-4* larvae compared to *TIR1; dpy-27::AID* larvae suggested that the X may be more derepressed in the strain also lacking *cec-4.* To test these hypotheses, we extracted total mRNA from synchronized L3 populations, which were grown in the presence of auxin starting at L1 or grown in the absence of auxin for controls. We performed these experiments at the L3 stage rather than at a later stage, because at this stage the delay in growth and development of the DPY-27 depleted strains is not yet apparent, and also to avoid complications arising from the contributions from the germline where the X chromosomes are not subjected to dosage compensation [44]. At the L3 stage, the germline consists of only about ~60 cells, and thus the RNA extracted from whole worms represents predominantly somatic tissues [45]. Three replicates were analyzed for each condition. PCA plots, hierarchical clustering, and Pearson correlation analyses were used to evaluate data quality (S8-S11 Figs). Since gene expression did not change much in some samples compared to WT, these samples did not necessarily cluster according to genotype. As discussed below, expression changes are minimal in *cec-4* mutants and in all samples without auxin treatment and this is reflected in the clustering and correlation analyses (S10-S11 Fig). We therefore also performed analysis on a more limited number of samples, the *TIR1; dpy-27::AID* and *TIR1; dpy-27::AID; cec-4* strains with and without auxin treatment (S8 and S9 Figs). The auxin treated samples do cluster away from non-auxin treated samples, and the *TIR1; dpy-27::AID; cec-4* strain treated with auxin clusters away from *TIR1; dpy-27::AID* strain treated with auxin (See PCA plot on S8A Fig).

To analyze X:A gene expression ratio, the mRNA-seq readout was plotted as a log2-fold change in gene expression on autosomes and the X in auxin-treated samples compared to controls (Figs 6, S12 and S8 File). Without exposure to auxin, the *dpy-27::AID* strain had similar gene expression levels to N2, although the slight X derepression was statistically significant. The minimal change in gene expression suggests that tagging DPY-27 with the degron tag did not significantly disrupt its function (S12A Fig). *cec-4* strains treated with auxin also displayed slight X depression compared to *cec-4* mutants without auxin exposure, indicating a minimal influence of auxin treatment on gene expression regulation (S12B Fig). The *TIR1; dpy-27::AID* and the *TIR1; dpy-27::AID; cec-4* strains also had a small degree of X derepression compared to wild type even in the absence of auxin, possibly due to somewhat leaky *TIR1* activity (S12C and S12D Fig), which is a phenomenon noted by other research groups as well [46–48]. Therefore, to assess the increase in X-linked gene expression resulting from the loss of DPY-27, we compared gene expression levels in the same strain with and without auxin and/or in auxin-treated strains with and without TIR1 expression (Fig 6). In comparison to *dpy-27::AID*, the strain *TIR1; dpy-27::AID* had significant upregulation of the X chromosome when treated with auxin, while autosomal gene expression remained unchanged (X derepression (defined as median X-linked log2 fold change – median autosomal log2 fold change) = 0.503, about 1.4-fold increase in gene expression) (Fig 6A). Similarly, auxin-treated *TIR1; dpy-27::AID* exhibited significant X derepression (X derepression = 0.435, about 1.35-fold

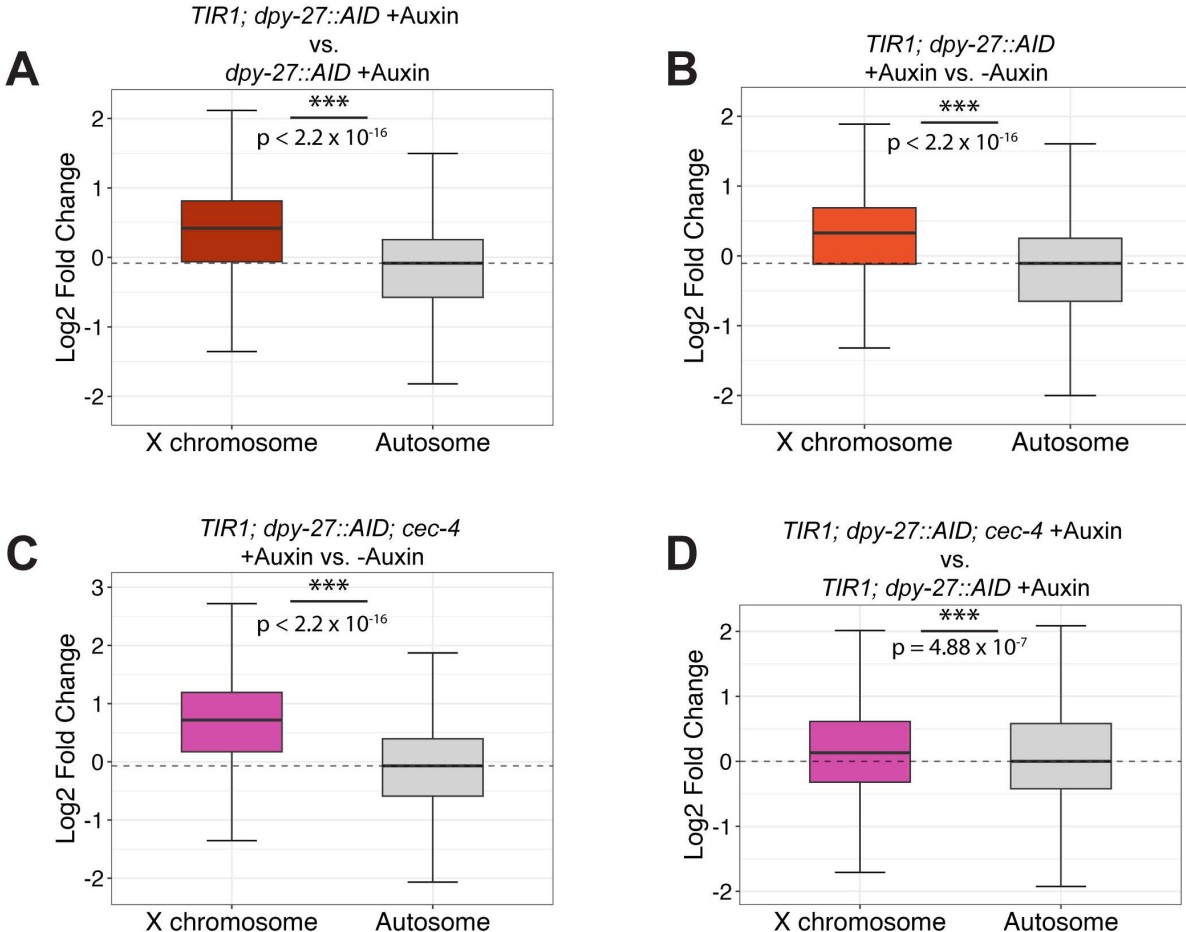

**Fig 6. The X-linked genes are derepressed in strains depleted of DPY-27.** (A-D) Boxplots depicting the distribution of the log2 fold change of X-linked genes and autosomal genes in strains being compared. A Wilcoxon rank-sum test was used to determine the statistical significance of the differential gene expression between the X and autosomes. (n.s. = not significant, * = p < 0.05, ** = p<0.01, *** = p < 0.001) (A) The log2 fold change in gene expression of the autosomes and the X chromosomes of *TIR1; dpy-27::AID* relative to the *dpy-27::AID* after both strains were exposed to auxin. (B)*TIR1; dpy-27::AID* treated with auxin compared to the same genotype grown in the absence of auxin. (C) *TIR1; dpy-27::AID; cec-4* treated with auxin was compared to no auxin treatment. (D) *TIR1; dpy-27::AID; cec-4* compared to *TIR1; dpy-27::AID,* both treated with auxin. P values for all comparisons are listed in S8 File.

increase in expression) compared to non-auxin-treated (Fig 6B). These results demonstrate that depletion of DPY-27 during the maintenance phase of dosage compensation leads to a significant increase in X-linked gene expression.

We then investigated whether the absence of CEC-4 further enhanced X derepression. Comparing the mRNA-seq gene expression profiles of *TIR1; dpy-27::AID; cec-4* treated with auxin to untreated populations of the same genotype, we observed a significant increase in X-linked gene expression (X derepression = 0.78, about 1.72-fold increase in gene expression) (Fig 6C). The degree of X derepression was greater than that caused by auxin treatment in *TIR1; dpy-27::AID* without the *cec-4* mutation (X derepression of 0.435). These results suggest that the loss of *cec-4* may exacerbate X dosage compensation defects. To confirm, we compared *TIR1; dpy-27::AID; cec-4* to *TIR1; dpy-27::AID* after auxin treatment of both genotypes. The difference between these two populations was substantial (X derepression = 0.13), suggesting that depletion of DPY-27 leads to a greater level of X derepression in a strain that lacks

CEC-4 (Fig 6D). These results are consistent with CEC-4 playing a supporting role in helping to maintain stable X chromosome repression in differentiated cells.

To investigate the influence of *cec-4* in more detail, we used chromosome coordinates of transcripts with changed gene expression to visualize the distribution of derepressed regions in each comparison (S13 Fig). CEC-4 binds to methylated H3K9 [32], a histone modification which is enriched on the arms of autosomes and the left end of the X chromosome [30]. These same genomic regions also make more frequent contacts with nuclear lamina proteins [30,31]. We therefore considered the possibility that genes whose expression changes due to the lack of *cec-4* would be preferentially located in the left end domain of the X chromosome. Consistent with a general defect in dosage compensation, in both the *TIR1; dpy-27::AID* and the *TIR1; dpy-27::AID; cec-4* auxin treated strains compared to no auxin controls, we observed a general derepression along the length of the X chromosomes (S13A and S13B Fig). Derepression was not more evident along the left end of the chromosome in the *TIR1; dpy-27::AID; cec-4* strain, suggesting that the impact of the *cec-4* mutation does not preferentially effect this chromosomal domain. When *cec-4* mutants were compared to WT, or when *TIR1; dpy-27::AID; cec-4* on auxin was compared to *TIR1; dpy-27::AID* on auxin (S13A and S13B Fig), similar conclusions were made. Gene expression differences were fairly evenly distributed in the genome, with no apparent preferential impact along H3K9me enriched domains. While somewhat surprising, these results are consistent with our previous study in which we found that the CEC-4's effect on the X chromosome is more chromosome-wide, and not limited to the left end domain [33] (see Discussion).

## Loss of *cec-4* activity exacerbates dosage compensation defects caused by a mutation in *dpy-21*

Our experiments showed that the loss of CEC-4 sensitizes hermaphrodite *C. elegans* to dosage compensation defects caused by DPY-27 depletion (Fig 6). To test whether loss of CEC-4 also exacerbates defects caused by other dosage compensation mutations, we sought to understand the interplay between the nuclear lamina protein CEC-4, and DPY-21, whose activity leads to X-specific enrichment of H4K20me1 [25]. Individually, the mutations in these pathways lead to phenotypes related to dosage compensation defects and changes in X-linked gene expression, but despite these changes, the mutants are viable [22,25,26,32,33]. Since *dpy-21* null mutants (*dpy-21(e428))* are viable [22], we performed these experiments with the null mutant rather than using the AID system. Therefore, our experiment tested the impact of these mutations throughout the lifespan of the animals, including both the establishment and the maintenance phases of dosage compensation. We constructed a double mutant *cec-4 (ok3124)* IV*; dpy-21(e428)* V strain (referred to as *cec-4; dpy-21).* The strain is viable but with visible defects, such as a Dpy phenotype.

We first evaluated the ability of hermaphrodites lacking functional *cec-4* and *dpy-21* to lay viable eggs. The total number of eggs laid by *cec-4; dpy-21* hermaphrodites was significantly smaller than all other strains observed, including the *dpy-21* and *cec-4* single mutants (Fig 7A). In all strains, most embryos hatched into larvae within 24 hours (Fig 7B). Most WT and *cec-4* mutant larvae developed into healthy adults. While the viability of *dpy-21* mutant larvae was somewhat reduced, it was not significantly different from the viability of *cec-4; dpy-21* double mutants (Fig 7C). Overall, these results suggest that the addition of the *cec-4* mutation does decrease the overall fitness of *dpy-21* mutants, most apparent in their ability to produce eggs.

To assess the level of X chromosome compaction, we used DPY-27 IF to mark the territories of the X chromosomes. DPY-27 maintains X localization in these mutants (S15 Fig), and previous studies have used DPY-27 staining in *cec-4* and *dpy-21* mutants to analyze X chromosome

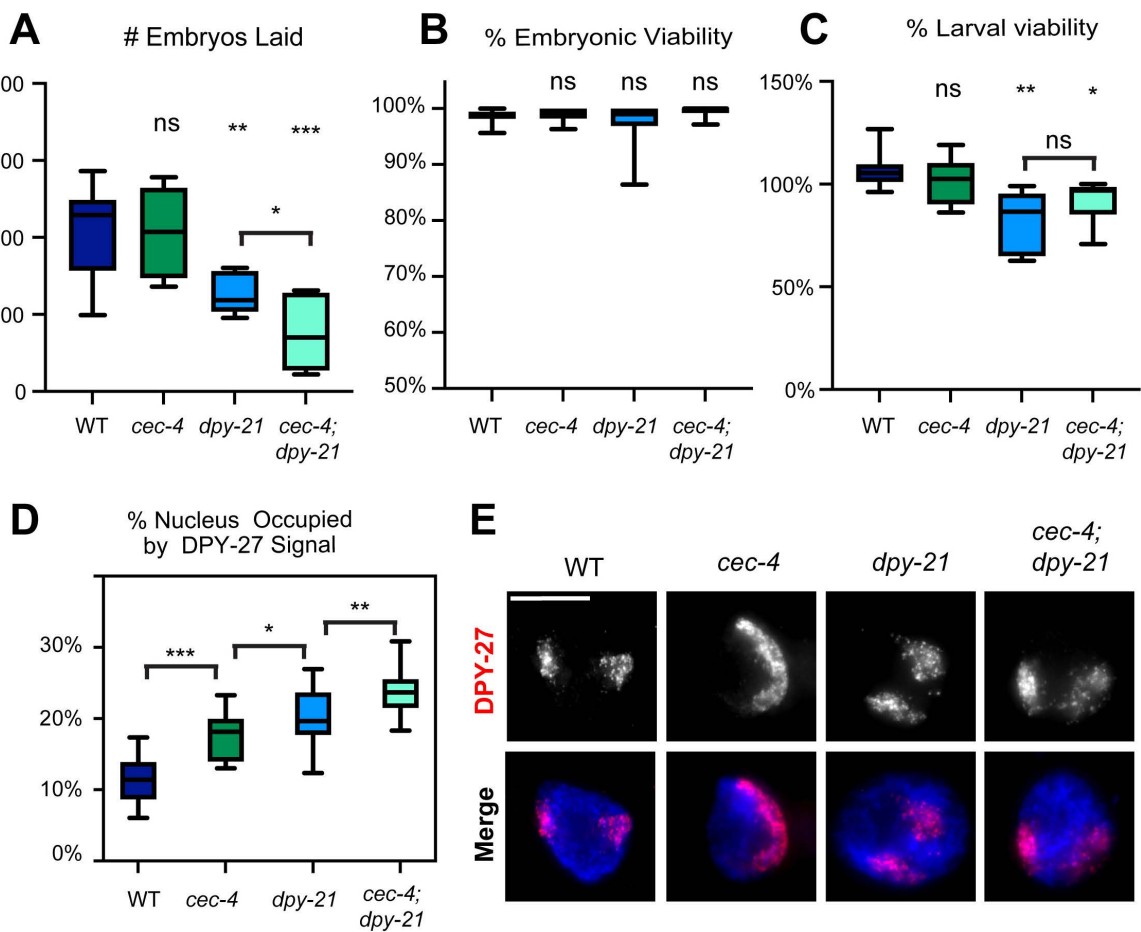

**Fig 7. Loss of CEC-4 exacerbates dosage phenotypes caused by mutations in *dpy-21*.** (A) The average of total number of eggs laid per worm in WT, *cec-4, dpy-21,* and *cec-4; dpy-21* backgrounds. (B) The average percentage of embryos that hatched after 24 hours. (C) The average percentage of larvae that survived to adulthood. (**A**-C) Progeny from 6-8 individual worms were scored per experiment for each genotype. Results shown are the combined averages from three independent experiments (18<n<24). Statistical significance was determined using Student's t-test. For statistical analysis and p values see S9 File. (D) The extent of X chromosome decompaction in WT, *cec-4, dpy-21,* and *cec-4; dpy-21* hermaphrodites. A total of 20 nuclei were analyzed from 3-5 independent experiments, no more than 2 nuclei per individual worm. Statistical significance was determined using Student's t-test. For statistical analysis and p values, see S10 File. (E) Representative images of adult hermaphrodite intestinal nuclei stained for DPY-27 (top) and merged images of DPY-27 (red) and DNA (DAPI, blue). Maximum intensity projections are shown. Unmodified images are shown on S14 Fig. Scale bar, 10 μm. n.s. = not significant, * = p < 0.05, ** = p<0.01, *** = p < 0.001.

compaction [25,49,33]. *cec-4* and *dpy-21* single mutants, as well as the *cec-4; dpy-21* double mutants all exhibited a prominently decondensed X chromosome (Fig 7E). Further quantitative analysis revealed that the X chromosomes were decondensed in *cec-4* and *dpy-21* single mutants, but in the *cec-4; dpy-21* double mutant strain the X occupied an even more significant proportion of the nucleus (Fig 7D and 7E). These observations show that adding the *cec-4* mutation to *dpy-21* mutants augments some defects associated with dosage compensation.

To determine the impact of these mutations on X chromosome gene repression, mRNA-seq was performed in L3 hermaphrodites (Fig 8, see S16 and S17 Figs for correlation and clustering analysis, and S8 File for statistical analysis). We previously reported subtle upregulation of the X chromosomes in *cec-4* mutant L1 larvae compared to the wild type [33]. In L3 larvae, this upregulation was even more modest and not statistically significant (X derepression = 0.027,

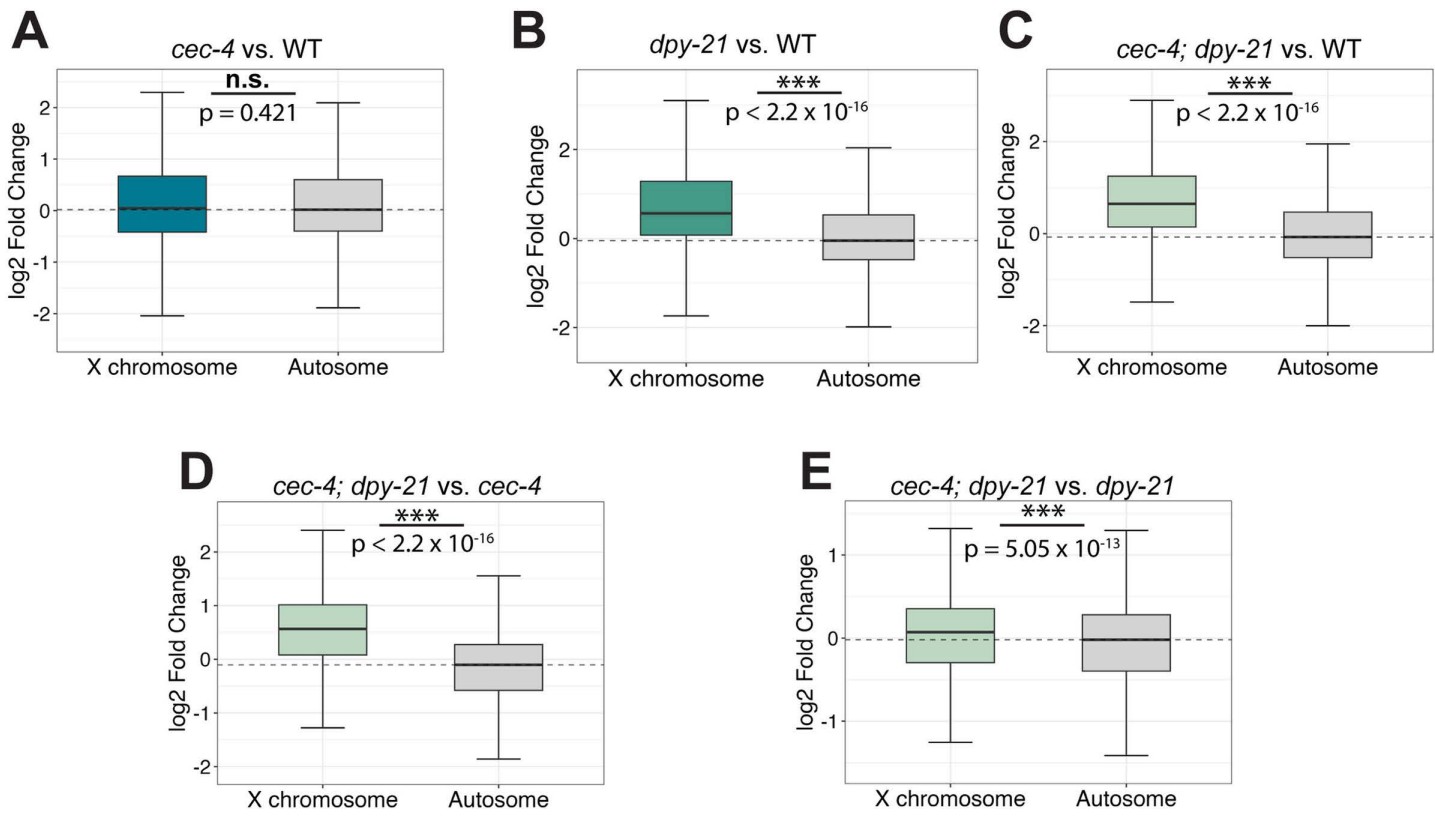

**Fig 8. Loss of CEC-4 exacerbates gene expression defects caused by mutations in *dpy-21*.** (A-E) Boxplots depicting the distribution of the expression change in X-linked genes and expression change in genes on autosomes is plotted as the log2 fold change of the strains being compared. Statistical significance is determined by the differences in gene expression between the X and autosomes by a Wilcoxon rank-sum test. (n.s. = not significant, * = p < 0.05, ** = p<0.01, *** = p < 0.001) (A) The log2 fold change in gene expression of the autosomes and the X chromosomes in *cec-4* mutant relative to the wild type (N2). (B) The log2 fold change in gene expression of the autosomes and the X chromosomes in *dpy-21* mutants relative to wild type (N2). (C) The log2 fold change in gene expression in *cec-4; dpy-21* relative to wild type (N2). (D) The log2 fold change in gene expression in *cec-4; dpy-21* compared to *cec-4*. (E) The log2 fold change in gene expression in *cec-4; dpy-21* compared to *dpy-21*. The derepression of the X chromosome genes is significantly higher in the double mutant than in single mutants.

p = 0.421) (Fig 8A), suggesting that lack of CEC-4 alone does not have a significant impact on X-linked gene regulation despite the observed changes in X chromosome compaction and nuclear localization. In contrast, the *dpy-21* mutant displayed a more pronounced derepression of the X chromosome (X derepression = 0.612, about 1.53-fold change, $p < 2.2 \times 10^{-16}$) (Fig 8B), consistent with previous results [25,29]. When we compared the *cec-4; dpy-21* double mutant to the wild type, we observed an even more drastic change in gene expression (X depression = 0.721, 1.65-fold change, $p = 2.2 \times 10^{-16}$) (Fig 8C). Furthermore, when juxtaposing the *cec-4; dpy-21* double mutant against its single mutant counterparts, the results underscored that *cec-4; dpy-21* expressed X-linked genes at a higher level than either single mutant (*cec-4; dpy-21* vs *cec-4*: X derepression = 0.669, $p = 2.2 \times 10^{-16}$, and *cec-4; dpy-21* vs *dpy-21*: X derepression = 0.09, $p = 5.05 \times 10^{-13}$) (Fig 8D and 8E). Given the relatively minor derepression of the X chromosome in *cec-4* mutants, a significant increase in the X-linked gene expression levels of *cec-4; dpy-21* compared to *cec-4* alone was expected (Fig 8D). The *dpy-21* single mutant already exhibited an upregulation in X chromosome gene expression; however, the *cec-4; dpy-21* double mutant significantly surpassed it in X derepression (Fig 8E). While *cec-4*, as a single mutation, does not substantially impact X-linked gene expression, our results show that the loss of *cec-4* sensitizes hermaphrodites to additional dosage compensation defects.

## Discussion

In this study, we investigated the relative importance of the DCC during two phases of dosage compensation: establishment and maintenance. We depleted DPY-27, a core protein subunit of the dosage compensation machinery, condensin I$^{DC}$, using the auxin-inducible degron system during embryonic and larval development. Our data indicated that DPY-27 function is essential during embryonic development, but the protein is dispensable for hermaphrodite survival during larval and adult stages. In an additional dosage compensation mechanism, the non-condensin DCC member DPY-21 significantly contributes to X-linked gene repression by enriching H4K20me1 on the X chromosome [25]. We show that the enrichment of H4K20me1 is lost rapidly after DPY-27 depletion, indicating that the mark must be continuously maintained, presumably by DPY-21 recruited to the X by DPY-27. In addition, after DPY-27 depletion, the X chromosomes are decondensed and more centrally located in the nucleus. These changes are accompanied by significant increases in X-linked gene expression. In addition to the contributions of the DCC, the nuclear lamina protein CEC-4 also contributes to X morphology and repression, potentially via its known role of tethering H3K9me3 chromatin to the nuclear periphery [33]. We found that a loss-of-function mutation in *cec-4* amplifies the sensitivity of hermaphrodites to the depletion/mutation of other dosage compensation processes. These results suggest that CEC-4 serves to stabilize repression initiated by condensin I$^{DC}$ or H4K20me1. By loss-of-function mutations in *cec-4* and *dpy-21* and depletion of DPY-27, we unexpectedly revealed that larval and adult hermaphrodites are resilient to depletion of known dosage compensation mechanisms and substantial increase in X-linked gene expression.

## Condensin I$^{DC}$ function is required for embryonic viability but not for larval and adult viability

A previous study using a cold-sensitive mutant for *dpy-27* demonstrated that inactivating DPY-27 during the comma stage of embryonic development leads to more extensive lethality than inactivation of the protein at later stages. However, the differences in viability counts at the different temperatures were only about 20% [39]. We re-examined this question using the auxin-mediated degradation system, where adding auxin leads to rapid and near complete depletion of DPY-27. Prior to auxin treatment, the strains had essentially 100% viability, while auxin treatment during embryogenesis led to essentially 100% lethality. These results demonstrate that condensin I$^{DC}$ activity is essential during embryonic development when cell fates are being specified, and cells begin to differentiate into different cell types. At the end of embryonic development, the embryo consists of about 500 cells, the majority of which have exited the cell cycle and begun terminal differentiation [38]. Depletion of condensin I after embryogenesis is not incompatible with viability despite developmental abnormalities in depleted larvae. This result raised two possibilities. First, it is possible that dosage compensation, once established, can be maintained in the absence of condensin I$^{DC}$. Second, it is possible that X chromosome repression is not maintained but the larvae can tolerate defects in X-linked gene regulation. Our results, discussed below, are consistent with the second scenario.

## Condensin I$^{DC}$ must remain on the X to maintain the condensed conformation

We show that compaction of the dosage compensated X cannot be maintained without DPY-27. By varying durations of auxin exposure and observing the condensin I$^{DC}$ subunit CAPG-1, we saw the loss of condensin I$^{DC}$ enrichment on the X and the gradual loss of CAPG-1 signal from the nucleus (Figs 2 and 3). We propose that the absence of DPY-27 leads to the

destabilization of the condensin I$^{DC}$ ring, resulting in the release of the complex from the X and gradual degradation of other complex members. The X chromosome also decondenses after condensin I$^{DC}$ loss, indicating that chromosome compaction must be actively maintained. Previous research investigating the maintenance of mitotic or meiotic chromosome architecture demonstrated that depletion or inactivation of mitotic condensin subunits after chromosome condensation has taken place leads to disorganized chromosome structure with reduced rigidity or a larger surface area [50–53]. Similar to what we observed about the continued need for condensin I$^{DC}$ to maintain the compact structure of dosage compensated X chromosomes, maintenance of mitotic chromosome architecture also requires the continued presence of condensin. However, in the studies of mitotic chromosomes, compaction was maintained after condensin loss, only the organization or rigidity of chromosomes was altered. In our study, compaction of the interphase X chromosome was not maintained. It has been suggested that during mitosis additional mechanisms, such as posttranslational histone modifications, contribute to chromosome condensation [54]. We hypothesize that in the absence of these mitosis-specific histone modifications in interphase, condensin function is necessary for compaction. We note that two posttranslational histone modifications associated with mitosis, namely enrichment of H4K20me1 and depletion of H4K16ac, are also features of interphase dosage compensated X chromosomes of *C. elegans* [26,27,54,56]. However, additional mitotic histone modifications that are not present on dosage compensated X chromosomes (for example H3S10Ph or depletion of additional acetylation marks) may also contribute to condensin independent chromosome compaction in mitosis [57]. Our findings indicate that the complete condensin complex must remain associated with the X for the maintenance of compaction during developmental stages in which most cells are post-mitotic. A recent study employed TEV protease-mediated cleavage of DPY-26, a subunit of condensin I$^{DC}$ to inactive the complex. Cleavage of DPY-26 in the L1 stage led to loss of TAD structures from the X, suggesting that condensin I$^{DC}$ is also required to maintain the fine-scale structural organization of the chromosome [58]. Compaction and organization of the chromosome into TADs are separable functions, as it is possible to disrupt TADs without loss of compaction [49]. Our results, together with the study by Das et al [58], suggest that the continued presence of condensin I$^{DC}$ is required to maintain both levels of chromosomal organization.

In the absence of both condensin I$^{DC}$ and CEC-4, the structure of dosage compensated X chromosomes is even more disrupted. However, the exact role of CEC-4 in the process remains unclear. We previously showed that the impact of CEC-4 on X chromosome compaction is most pronounced in the middle gene-rich domain, and the left arm the chromosome (where the H3K9me3 mark and nuclear lamina interactions are enriched) remains relatively unaffected [33]. Gene expression changes on the X were also not limited to the H3K9me3-enriched domain (to which CEC-4 binds) both in our previous study [33] and our current one (S13 Fig). Thus, the effect appears more chromosome-wide. This is consistent with previous studies that showed a lack of correlation between DCC binding site on the X and genes being regulated by dosage compensation [24,59], and a lack of correlation between DCC-induced TAD formation and changes in gene expression [19]. Thus, it has been suggested that dosage compensation impacts gene expression more in a chromosome-wide manner, rather than locally by regulating individual genes.

## The presence of DPY-27 is required to actively maintain the enrichment of H4K20me1 on the X

Shortly after the loss of DPY-27 signal, the H4K20me1 mark exhibits a loss of specific localization on the X with increasing auxin exposure time (Figs 2 and 4). We propose that the continued presence of DPY-21 (presumable recruited by DPY-27) on the X is necessary to

maintain H4K20me1 enrichment on the X. Notably, the presence of DPY-21 is not essential for the localization of condensin I$^{DC}$ on the X; however, for DPY-21 to bind the X, the presence of condensin I$^{DC}$ is required [22]. Our results are similar to what was seen for H3K9me2 and MET-2. Maintenance of this chromatin mark in *C. elegans* also requires the continued presence of the enzyme that places it [60]. Our results suggest that functional condensin I$^{DC}$ plays a role in maintaining the localization of DPY-21 on the X, and its continued presence is required even in post-mitotic developmental stages. We hypothesize that in the absence of DPY-27 and DPY-21, either there is frequent enough histone turnover to lose the H4K20me1 enrichment or that the X is rendered accessible to histone modifiers that remove the H4K20me1 or modify the mark to di- or trimethylation.

## Condensin I$^{DC}$ is required to maintain X-linked gene repression in hermaphrodites

Severe depletion or mutation of dosage compensation in *C. elegans* is hermaphrodite lethal. Unexpectedly, we observed that hermaphrodites survive as larvae with near complete depletion of DPY-27, albeit with major phenotypic defects (Fig 1C)**.** Furthermore, they survive without other vital contributors to X-linked gene regulation in addition to DPY-27 depletion, namely, the activities of CEC-4 and H4K20me1 (Figs 1 and 4). In short, auxin-treated *TIR1; dpy-27::AID; cec-4* larvae survive without measurable activity of any known dosage compensation mechanism. Our study revealed increases in X-linked gene expression beyond levels reported in *dpy-27* null mutant embryos or L1 larvae (log2 fold change in X-linked gene expression of less than 0.3) [29] or *sdc-2* mutant embryos treated with *sdc-2* RNAi (log2 fold change about 0.6) [49]. Dosage compensation defects in these mutants lead to lethality. Our most significant increase in X-linked gene expression was measured in *TIR1*; *dpy-27::AID; cec-4* L3 larvae after auxin exposure with a log2 fold change in X-linked gene expression at a value of 0.78 (Fig 5). These larvae survive despite more significant gene expression defects than previously reported for embryos dying of dosage compensation defects. We suggest that the absence of these dosage compensation processes is acceptable in these animals because the continued repression of the X is no longer essential after early development once cell fates have been specified and the cells have differentiated.

In mice, the critical contributor to dosage compensation, the long non-coding RNA *Xist*, can be deleted with minimal change to X repression as long as it is deleted during the maintenance phase after X chromosome inactivation is fully established [61,62]. However, unlike the continued repression of the mouse X despite the loss of *Xist*, we observed a significant derepression of X-linked genes in *C. elegans* hermaphrodites after the depletion of DPY-27. Derepression was also observed in a recent study after inactivation of condensin I/I$^{DC}$ during the L1 stage [58]. The difference between dosage compensation maintenance in mice and *C. elegans* might be due to the nature of the repressive mechanisms. In mice, X chromosome inactivation involves the establishment of facultative heterochromatin on the X chromosomes, including DNA methylation and heterochromatic histone modifications [63]. Studies in other organisms have shown that these chromatin marks can be maintained by propagating them during the S phase [64–67], even without the initial trigger. The repressive mechanisms involved in *C. elegans* dosage compensation are different. H4K20me1 patterns are not propagated during DNA replication; in fact, H4K20me1 levels decrease in S phase due to the activity of an H4K20me1 demethylase PHF8 [68]. Instead, H4K20me1 is thought to be established during mitosis [55] and then converted to H4K20me2/3 methylation after mitotic exit [55]. Our results suggest that maintenance of this mark on dosage compensated X chromosomes requires the continued presence of DPY-27, presumably recruiting the H4K20me2 demethylase DPY-21 [25].

## The loss of function of *cec*-4 sensitizes hermaphrodites to the loss of DPY-27 or DPY-21

The *cec-4* mutation, combined with DPY-27 depletion, showed only minor changes in nuclear compartmentalization and chromosome compaction compared to DPY-27-depleted strains with functional CEC-4 (Fig 5). However, there was a notable decrease in hermaphrodite viability of DPY-27 depleted larvae with a *cec-4* mutation compared DPY-27 depleted larvae without a *cec-4* mutation (Fig 1E), suggesting disruption to X-linked gene regulation approaching unacceptable limits for survival. When a *cec-4* mutation was combined with a *dpy-21* mutation, there was an observable increase in X decompaction and a decrease in brood size in the double mutant compared to single mutants, again suggesting greater disruption to dosage compensation (Fig 7). mRNA-seq analysis indicated that the loss-of-function mutation of *cec-4* alone minimally increased X-linked gene expression (Fig 8A). However, in scenarios where a *cec-4* mutation was coupled with DPY-27 depletion or *dpy-21* mutation, we observed a substantial increase in X-linked gene expression, significantly more than in strains without the *cec-4* mutation (Figs 6 and 8). Previous studies have highlighted the involvement of CEC-4 in cellular differentiation and stabilizing cell fates by anchoring heterochromatic regions of the genome to the nuclear lamina [32]. We propose that the same activity may also stabilize repression of the X chromosomes by the dosage compensation complex. Cells actively undergoing differentiation demand precise gene expression levels. However, we hypothesize that the loss of some repressive mechanisms after differentiation are acceptable, partly because CEC-4 may help lock down established gene expression patterns, including the repressed state of dosage-compensated X chromosomes. It should be noted, however, that the loss of *cec-4* challenges the health of *C. elegans* larvae depleted of other dosage compensation proteins, but most of the larvae survive. We hypothesize that losing these mechanisms, including CEC-4 function, at the larval developmental stage does not result in outright lethality because cellular differentiation is more complete.

While DCC-mediated mechanisms are crucial for viability during the establishment of dosage compensation, our results demonstrate that they are not essential for survival during the maintenance stage. During larval and adult developmental stages, hermaphrodites survive

**Table 1. *C. elegans* strains.**

| Genotype | Strain | Source |
|---|---|---|
| Wild Type | Bristol N2 | CGC |
| *dpy-21(e428)* V | EKM71 | This study |
| *cec-4 (ok3124)* IV | EKM121 | This study |
| *cec-4 (ok3124)* IV; *dpy-21(e428)* V | EKM222 | This study |
| ieSi57[Peft-3::*TIR1*::mRuby::unc-54 3'UTR, cb-unc-119(+)] II; *dpy-27::AID::MYC* (xoe41) III | EKM237 | This study |
| *dpy-27::AID::MYC* (xoe41) III | EKM227 | [41] |
| *dpy-27::AID::MYC* (xoe41) III; *cec-4 (ok3124)* IV | EKM233 | This study |
| ieSi57[Peft-3::*TIR1*::mRuby::*unc-54* 3'UTR, *cb-unc-119*(+)] II; *dpy-27::AID::MYC*(xoe41) III; *cec-4*(ok3124) IV | EKM238 | This study |
| wrdSi3 [*sun-1*p::*TIR1*::F2A::mTagBFP2::AID*::NLS::tbb-2 3'UTR] II; *dpy-27*(xoe41[*dpy-27::AID::myc*]) III; *him-8* (*me4*) IV | JEL1197 | [41] |
| *unc-119* (*ed3*); ieSi57 [*Peft-3*::*TIR1*::mRuby::*unc-54* 3'UTR, *cb-unc-119*(+)] II | CA1200 | [42] |

without any known dosage compensation activity. This resilience raises questions about the necessity of continued X repression after embryonic development.

## Materials and methods

### Caenorhabditis elegans strains

The strains used are shown in Table 1. Strains were maintained at 20°C on nematode growth media (NGM) plates seeded with the *Escherichia coli* strain OP50 as a food source, as described in [69]. For some experiments requiring larger number of worms, strains were grown on high-growth media (HGM) plates were seeded with *E. coli* strain NA22, as described in [70].

The alleles for *dpy-21 (e428)* and *cec-4 (ok3124)* were initially acquired from the *Caenorhabditis* Genetics Center (CGC) as the strains CB428 and RB2301, respectively. Each strain was backcrossed to Bristol N2 six times to generate the strains EKM71 *dpy-21 (e428)* and EKM121 *cec-4 (ok3124)*. EKM222 was obtained by crossing EKM71 to EKM121.

The strain JEL1197 was received from and generated by [41]. To isolate the *dpy-27::AID::MYC* (xoe41) allele, JEL1197 was crossed to wild type to produce EKM227. EKM227 was then crossed to CA1200 [42] to create EKM237.

To add *cec-4 (ok3124)* to the background, EKM227 was crossed with EKM121. The subsequent strain, EKM233, was crossed with CA1200 to generate EKM238.

### Antibodies

Antibodies used in this study include rabbit anti-H4K20me1 (Abcam ab9051), rabbit anti-DPY-27 [11], rabbit and goat anti-CAPG-1 [33], and rat anti-HTZ-1 [71]. Secondary anti-goat, anti-rat, and anti-rabbit antibodies were purchased from Jackson Immunoresearch.

### Synchronized worm growth and collection

To obtain synchronized worm populations, gravid adults were bleached to collect embryos [72]. During bleaching, the hermaphrodites degrade and leave only embryos, which are then collected. After washing with sterile 1X M9 salt solution [73], the embryos are isolated and shaken slowly overnight in a flask containing M9 buffer to allow them to hatch. Synchronized L1s were plated onto either NGM with OP-50 or HGM with NA22 plates as needed. Synchronized L1s were used in RNA-seq, IF, and larval development assays. After synchronization, L1s were grown to L3, or adult as needed per assay.

### Auxin Treatment

Strains were subjected to auxin (Thermo Fisher Alfa Aesar Indol-3-Acetic Acid #A10556) by incorporation into NGM or HGM plates at a final concentration of 4mM, as described in [42,74]. Auxin-containing plates were maintained in the dark at 4°C for up to a month.

### Assay of viability upon auxin exposure initiated during embryogenesis

L4 hermaphrodites were placed on an auxin-containing plates or NGM plates for controls to ensure all embryonic development occurs while exposed to auxin. Plates were maintained in the dark at 20°C. Worms were placed on plates with or without auxin for an egg-laying period of 24 hours. After this period, the adult was removed from the plate, and the embryos laid were counted. 24 hours later, the number of dead larvae and dead embryos were quantified. Live adults were counted 3 days after the initial egg-laying period. Three independent replicates were performed. Differences were evaluated using Fisher's exact test comparing the numbers of dead and live progeny for each genotype.

### Assay of viability upon auxin exposure initiated during larval development

Experiments testing larval development began with synchronized L1 larvae. The synchronized population was resuspended in M9 and split evenly between an auxin plate and a plate without auxin for 3 days. Worms surviving after 3 days were counted. Assuming an equal number of larvae initially plated, the final counts are reported as a ratio of survival on auxin to survival off auxin. Three independent replicates were performed. Differences were evaluated using Chi square test comparing the number of surviving worms on auxin to the number surviving off auxin of each genotype.

### Brood Count Assay

Young adult hermaphrodites were allowed to lay eggs on NGM and moved to new plates each day until fertilized embryos were no longer produced. For each plate, after removal of the parent, the number of embryos laid were counted, and after 24 hours, the number of dead embryos were counted. After 3 days of development, the number of surviving adults were counted. Embryonic viability was quantified as the number of hatched embryos/total embryos. Larval viability was quantified as total number of adults/hatched embryos.

### Immunofluorescence (IF)

Hermaphrodite worms were dissected in 1x sperm salts (50 mM Pipes pH 7, 25 mM KCl, 1 mM MgSO$_4$, 45 mM NaCl, and 2 mM CaCl$_2$) and fixed in 4% paraformaldehyde (PFA) diluted in 1X sperm salts on a glass slide for 5 minutes in a humid chamber containing PBS-T (PBS with 0.1% and Triton X-100). Slides were frozen on dry ice for at least 20 minutes with a 20x20 mm coverslip over the diluted PFA and dissected *C. elegans*. Using a razor blade, the coverslip was removed by flicking the coverslip off the slide. The slides were then washed in PBS-T 3 times for 10 minutes each. 30 μL of diluted primary antibody was applied to each slide, a piece of parafilm was placed over the spot with the antibody to slow evaporation. The slides were incubated in a humid chamber at room temperature overnight. Slides were washed in PBS-T 3 times for 10 minutes and stained with corresponding secondary antibodies diluted 1:100 in PBS-T at 37°C for 1 hour. Three more washes in PBS-T were completed after the secondary antibody incubation. In the third wash, the nucleus was stained using DAPI diluted in PBS-T for 10 minutes at room temperature. Slides are mounted in Vectashield (Vector Labs).

### Fluorescence *in situ* hybridization (FISH)

Degenerate Primer PCR was used to amplify purified yeast artificial chromosomes (YACs) containing sequences corresponding to regions of the X chromosome and then labeled with dCTP-Cy3 alongside standard nucleotides using random priming as in [33,75]. The X paint FISH probe generated this way covers about 90% of the X. Adult hermaphrodites were dissected on glass slides in 1x sperm salts and fixed in 4% PFA. After 5 minutes of fixation, the specimen was covered with a 20x20 coverslip and placed on dry ice for at least 20 minutes. The coverslip was then flicked off the slide with a razor blade. Slides were washed in PBS-T three times and subjected to an ethanol series of increasing concentrations (70%, 80%, 90%, 100% EtOH) for 2 minutes each, after which the slides were air dried. To each slide, 10uL of X paint probe was added and incubated at 37°C overnight. Next day, the slides were washed in a 39°C water bath in 2x SSC/50% formamide three times, in 2X SSC solution three times for 5 minutes each, then in 1x SSC once for 10 minutes, followed by a room temperature wash in 4x SSC. DAPI was included in the final 4x SSC wash and slides were mounted using Vectashield. For FISH followed by IF, DAPI was excluded in the final 4X SSC wash, after which IF was performed as described above.

## Microscopy

Microscopy was performed using an Olympus BX61 microscope and a 60X APO oil immersion objective. Images were taken with a Hamamatsu ORCA-ER High-Resolution Monochrome Cooled CCD (IEEE 1394) camera. Images of nuclei were captured using the Slidebook 5 program (Intelligent Imaging Innovations). Only nuclei with a clear separation from the rest of tissue were imaged, no more than 3 per worm, and often no more than 1. All experiments included imaging nuclei from at least three independent experiments. A nucleus from the set 20 to 50 nuclei was chosen as a representative image. All representative images shown are maximum intensity projections of image stacks taken with 0.2-micrometer Z-spacing. Quantification (see below) was performed on the 3D stacks or individual Z planes, not on the maximum intensity projection images. For images used for fluorescence intensity quantification, exposure times were standardized to wild type. All images in the same type of analysis were then subjected to identical min-max adjustments.

## Quantification of signal intensity and signal enrichment

Fluorescence intensity of the DPY-27, CAPG-1, and H4K20me1 signals was quantified in Slidebook 5. For each nucleus, the segment mask function was used to manually capture the DAPI signal in 3D space and to create a DAPI mask. The mean fluorescence for the appropriate channel (DPY-27, CAPG-1, H4K02me1) in the DAPI mask in the 3D image stacks was recorded. This value was then divided by mean fluorescence signal of the same channel in the same image in a region that represented background fluorescence. Thus, the signal intensity represents mean signal intensity in the nucleus/background intensity.

To quantify enrichment of the signal, the line intensity function was used in Slidebook on a single Z plane from the middle of the nucleus. A line was drawn across the nucleus including the enriched region (presumably the X chromosome). In nuclei with no clear enrichment after DPY-27 depletion, a random line was drawn across the nucleus. The peak signal intensity on the line-scan was divided by the lowest signal intensity to obtain the ratio of relative enrichment of CAPG-1 or H4K20me1 on the X chromosomes over the rest of the nucleus.

## Quantification of X Volume

Z-stacked images of intestinal nuclei of hermaphrodites were trimmed to 10-15 slices at to include only planes with nuclear signal. Background signal was subtracted in ImageJ Fiji with a rolling ball radius of 50 pixels. After splitting the channels, the 3dManager from the 3dSuite plugin was used to create a three-dimensional ROI of the DAPI signal and a three-dimensional ROI of the FISH signal in the Cy3 channel [76]. For X volume quantification, the area of the Cy3 channel ROI inside the DAPI ROI was then compared in the 3dSuite 3DManager to find the X chromosome volume in the nucleus. The mask for the X chromosomes was limited to the area that overlaps with the DAPI mask signal, ensuring that the quantifications do not include non-specific staining of X paint FISH. The proportion of the nucleus occupied by the X is a ratio of the X signal volume to the DAPI signal volume measured in voxels (volumetric pixels).

## Three-zone assay

From the images of nuclei stained using FISH targeting the X chromosome, the middle slice with the most in-focus FISH signal was selected. The background was subtracted in ImageJ FIJI with a rolling ball radius of 50 pixels. The DAPI and Cy3 channels were then split and thresholded. Using the native ROI manager in ImageJ FIJI, the threshold of the DAPI mask was used to draw three concentric ellipses that formed three zones of equal area in the

nucleus. The Cy3 signal overlap with each zone was calculated using the ROI Manager to return the X chromosome FISH signal percentage in each zone. For this assay, only nuclei with smooth edges and an elliptical shape were used.

## mRNA-seq

Synchronized L1s were placed on HGM plates with or without auxin and NA22 [77]. 24-hours later, L3 larvae were collected in M9. A small number of the larvae were removed and allowed to grow to adulthood for control staining experiments to ensure complete DPY-27 depletion. After pelleting, excess M9 was removed, and the remaining pellet was snap-frozen in liquid nitrogen with 1mL of Trizol. RNA was extracted and cleaned from the collected *C. elegans* using Qiagen RNeasy Kit. Poly-A enrichment, library prep, and next-generation sequencing were conducted in the Advanced Genomics Core at the University of Michigan. RNA was assessed for quality using the TapeStation or Bioanalyzer (Agilent). Samples with RINs (RNA Integrity Numbers) of 8 or greater were subjected to Poly-A enrichment using the NEBNext Poly(A) mRNA Magnetic Isolation Module (NEB catalog number E7490). NEBNext Ultra II Directional RNA Library Prep Kit for Illumina (catalog number E7760L) and NEBNext Multiplex Oligos for Illumina Unique dual (catalog number E6448S) were then used for library prep. The mRNA was fragmented and copied into first-strand cDNA using reverse transcriptase and random primers. The 3' ends of the cDNA were then adenylated, and adapters were ligated. The PCR products were purified and enriched to create the final cDNA library. Qubit hsDNA (Thermofisher) and LabChip (Perkin Elmer) checked the final libraries for quality and quantity. The samples were pooled and sequenced on the Illumina NovaSeqX 10B paired-end 150 bp, according to the manufacturer's recommended protocols. BCL Convert Conversion Software v4.0 (Illumina) generated de-multiplexed Fastq files. The reads were trimmed using CutAdapt v2.3. FastQC v0.11.8 was used to ensure the quality of data [78,79]. Reads were mapped to the reference genome WBcel235 and read counts were generated using Salmon v1.9.0 [80]. Differential gene expression analysis was performed using DESeq2 v1.42.0 [81]. Downstream analyses were performed using R scripts and packages. For visualizing gene expression changes along chromosomes, log2 fold change data in each genotype comparison was displayed using the Chromomap package [82] after filtering for all reads at a significance level of $p<0.05$. Alternatively, heatmaps were constructed displaying log2 fold change in gene expression arranged by chromosomal coordinates. Reads were positioned according to the chromosomal coordinates included in annotations provided by the reference genome in the previous RNAseq analysis and were displayed according to the *C. elegans* chromosome lengths found on WormBase.

## Supporting information

**S1 Fig. Unmodified original images for** Fig 2. **(A)** Unmodified original images of panels shown on Fig 2A. DPY-27 IF is shown in red, DNA stain (DAPI) in blue. **(B)** Unmodified original images of panels shown on Fig 2B. DPY-27 IF is shown in red, HTZ-1 IF in green (used as staining control) and DNA stain (DAPI) in blue. The main figure only shows that DPY-27 and DAPI color channels. Scale bars, 10 μm.
(TIF)

**S2 Fig. Reproducibility of DPY-27 depletion.** Control non-auxin treated worms and worms treated with auxin from the L1 stage from 3 days, or from day 1 of adulthood for 1 hr, or 24 hours were dissected and stain with anti-DPY-27 antibody, and anti-HTZ-1for staining control. Results from three independent experiments are shown. The merged images show DPY-27 in red and DNA (DAPI) in blue. Scale bars, 10 μm.
(TIF)

**S3 Fig. Unmodified original images for panels shown on S2 Fig.** DPY-27 is shown in red, HTZ-1 (staining control) in green, and DNA (DAPI) in blue. Scale bars, 10 μm.
(TIF)

**S4 Fig. Unmodified original images for panels shown on** Fig 3. **(A)** Unmodified images for Fig 3A. CAPG-1 staining is shown in green, H4K20me1 staining in red, DNA (DAPI) in blue. The main figure only shows the CAPG-1 and DAPI channels. **(B)** Unmodified images for Fig 3B. CAPG-1 staining is shown in green, HTZ-1 (staining control) in red, and DNA (DAPI) in blue. The main figure only shows the CAPG-1 and DAPI channels. Scale bars, 10 μm.
(TIF)

**S5 Fig. Unmodified original images for panels shown on** Fig 4. **(A)** Unmodified images for Fig 4A. CAPG-1 staining is shown in green, H4K20me1 staining in red, DNA (DAPI) in blue. The main figure only shows the H4K20me1 and DAPI channels. **(B)** Unmodified images for Fig 4B. H4K20me1 staining is shown in green, HTZ-1 (staining control) in red, and DNA (DAPI) in blue. The main figure only shows the H4K20me1 and DAPI channels. Scale bars, 10 μm.
(TIF)

**S6 Fig. Unmodified original images for panels shown on** Fig 5. X chromosome paint FISH probe is shown in red and DNA (DAPI) in blue. Scale bars, 10 μm.
(TIF)

**S7 Fig. Three-zone assay of X localization within the nucleus with and without auxin treatment.** (**A**) Three-zone assay for the X paint FISH signal in strains grown without auxin (-) or after auxin treatment from L1 larval stage to adulthood (L) showing all three concentric zones. Fig 5 shows only the data for the innermost zone. (**B**) The proportion of the X paint signal seen in the peripheral zone of the nucleus. Differences between samples were evaluated using unpaired Student's t-test. For complete statistical analysis see **S7 File**. n.s. = not significant, * = p < 0.05, ** = p<0.01, *** = p < 0.001.
(TIF)

**S8 Fig. Analysis of *dpy-27::AID* RNA-seq samples. (A)** Principal component analysis (PCA) plot depicting the relationships among *TIR1; dpy-27::AID*, and *TIR1; dpy-27::AID; cec-4* with and without auxin treatment. **(B)** Unsupervised hierarchical clustering of *TIR1; dpy-27::AID*, and *TIR1; dpy-27::AID; cec-4* with and without auxin treatment using Manhattan distance to calculate distance between samples, and ward.D2 algorithm to cluster samples.
(TIF)

**S9 Fig. Pearson correlation analysis of *dpy-27::AID* RNA-seq samples.** Heatmap represents color corresponding to the Pearson correlation coefficient between any two samples. Samples analyzed included the *TIR1; dpy-27::AID*, and *TIR1; dpy-27::AID; cec-4* strains with and without auxin treatment.
(TIF)

**S10 Fig. Clustering analysis of all RNA-seq samples.** Unsupervised hierarchical clustering of all RNA-seq using Manhattan distance to calculate distance between samples, and ward.D2 algorithm to cluster samples.
(TIF)

**S11 Fig. Pearson correlation analysis of all RNA-seq samples.** Heatmap represents color corresponding to the Pearson correlation coefficient between any two samples.
(TIF)

**S12 Fig. mRNA-seq analysis of dpy-27::*AID* strains. (A-D)** Boxplots depicting the distribution of the expression difference of X-linked genes and expression difference of genes on autosomes. Gene expression is plotted as the log2 ratio of the strains being compared. Statistical significance is determined by the differences in gene expression between the X and autosomes by a Wilcoxon rank-sum test. Samples compared are shown above the boxplots (n.s. = not significant, * = p < 0.05, ** = p<0.01, *** = p < 0.001).
(TIF)

**S13 Fig. Analysis of gene expression change in strains with and without *cec*-4 mutations mapped onto chromosomal regions. (A)** Chromomap analysis of log2 fold change in gene expression along the X chromosomes and autosomes. Samples compared are indicated above the maps. **(B)** Heatmap of log2 fold changes along each chromosome arranged by chromosomal coordinates. Comparisons are indicated at the bottom.
(TIF)

**S14 Fig. Unmodified original images for panels shown on Figs 7 and S15. (A)** Original unmodified images for Fig 7E. DPY-27 IF staining is shown in red and DNA (DAPI) in blue. **(B)** Original unmodified images for S15 Fig. DPY-27 IF is shown in green, X chromosome paint FISH probe is shown in red and DNA (DAPI) in blue. Scale bars, 10 μm.
(TIF)

**S15 Fig. DPY-27 localizes to the X chromosomes in *cec*-4 and *dpy*-21 mutants** . Signal for DPY-27 IF colocalizes with X-paint FISH signal. In the merged image, DPY-27 is shown in blue and X paint FISH signal in red. DAPI is shown in grayscale. Scale bars, 10 μm.
(TIF)

**S16 Fig. RNA-seq analysis of *cec*-4 and *dpy*-21 mutants. (A)** Principal component analysis (PCA) plot depicting the relationships among wild-type, single mutants (*cec-4* and *dpy-21*), and double mutants (*cec-4; dpy-21*) based on gene expression profiles. **(B)** Unsupervised hierarchical clustering of *cec-4* and *dpy-21* mutant RNA-seq datasets using Manhattan distance to calculate distance between samples, and ward.D2 algorithm to cluster samples.
(TIF)

**S17 Fig. Pearson correlation analysis of *cec*-4 and *dpy*-21 mutant data sets** . Heatmap represents color corresponding to the Pearson correlation coefficient between any two samples.
(TIF)

**S1 File. Numerical data and statistical analysis for Fig 1B.** Embryonic and larval viability after auxin exposure starting with the L4 stage of the parent. Total counts from three independent replicates are shown. Statistical analysis was performed using Fisher's exact tests. Statistically significant differences (p<0.05) are highlighted yellow. NA, not applicable.
(XLSX)

**S2 File. Numerical data and statistical analysis for Fig 1E.** Survival of larvae exposed auxin from the L1 stage was quantified. Total counts from three independent experiments are shown. Statistical analysis was performed using Chi square tests. Statistically significant differences (p<0.05) are highlighted yellow.
(XLSX)

**S3 File. Numerical data and statistical analysis for Fig 2C.** Quantification of DPY-27 fluorescence intensity. Statistical analysis was done using Brown-Forsythe and Welch ANOVA tests with multiple comparisons. Statistically significant differences (p<0.01) are highlighted yellow.
(XLSX)

**S4 File. Numerical data and statistical analysis for** Fig 3C **and** 3D**.** Quantification of CAPG-1 enrichment and fluorescence intensity. Statistical analysis was done using Brown-Forsythe and Welch ANOVA tests with multiple comparisons. Statistically significant differences (p<0.05) are highlighted yellow.
(XLSX)

**S5 File. Numerical data and statistical analysis for** Fig 4C **and** 3D**.** Quantification of H4K20me1 enrichment and fluorescence intensity. Statistical analysis was done using Brown-Forsythe and Welch ANOVA tests with multiple comparisons. Statistically significant differences (p<0.05) are highlighted yellow.
(XLSX)

**S6 File. Numerical data and statistical analysis for** Fig 5B**.** Analysis of X chromosome decondensation. Statistical analysis was done using Student's t-tests. Statistically significant differences (p<0.05) are highlighted yellow.
(XLSX)

**S7 File. Numerical data and statistical analysis for** Fig 5C**.** Analysis of X chromosome localization within the nucleus via the three-zone assay. Statistical analysis was done using Student's t-tests. Statistically significant differences (p<0.05) are highlighted yellow.
(XLSX)

**S8 File. Statistical analysis of mRNA-seq.** p values for the indicated comparisons are shown. Statistically significant differences (p<0.05) are highlighted yellow.
(XLSX)

**S9 File. Numerical data and statistical analysis for** Fig 7A**,** 7B **and** 7C**.** Progeny viability in *dpy-21* and *cec-4* mutants. Average progeny counts from individual worms (6<n<8), collected over three independent experiments are shown. Statistical differences were evaluated using Student's t-tests. Statistically significant differences (p<0.05) are highlighted yellow.
(XLSX)

**S10 File. Numerical data and statistical analysis for** Fig 7D**.** Analysis of X chromosome decond(XLSX)ensation in *cec-4* and *dpy-21* mutants. Quantification of a total of 20 nuclei collected over the course of 3 to 4 independent experiments are shown. Statistical significance was evaluated using Student t-tests. Statistically significant differences (p<0.05) are highlighted yellow.
(XLSX)

## Acknowledgments

We thank Sarah VanDiepenbos, Hend Almunaidi, and Lillian Tushman for their contributions to a preliminary analysis of the *cec-4 and dpy-21* mutants, as well as all members of the laboratory for helpful discussions. We would like to thank the laboratory of Dr. JoAnne Engebrecht for sharing the JEL1197 strain [41].

## Author contributions

**Conceptualization:** Jessica Trombley, Audry I Rakozy, Eshna Jash, Györgyi Csankovszki.

**Data curation:** Christian A McClear, Eshna Jash, Györgyi Csankovszki.

**Formal analysis:** Jessica Trombley, Audry I Rakozy, Christian A McClear, Eshna Jash, Györgyi Csankovszki.

**Funding acquisition:** Györgyi Csankovszki.

**Investigation:** Jessica Trombley, Audry I Rakozy, Christian A McClear, Eshna Jash, Györgyi Csankovszki.

**Methodology:** Jessica Trombley, Audry I Rakozy, Christian A McClear, Eshna Jash, Györgyi Csankovszki.

**Project administration:** Györgyi Csankovszki.

**Supervision:** Györgyi Csankovszki.

**Validation:** Jessica Trombley, Audry I Rakozy, Christian A McClear, Eshna Jash, Györgyi Csankovszki.

**Visualization:** Jessica Trombley, Audry I Rakozy, Christian A McClear, Eshna Jash, Györgyi Csankovszki.

**Writing – original draft:** Jessica Trombley, Györgyi Csankovszki.

**Writing – review & editing:** Jessica Trombley, Eshna Jash, Györgyi Csankovszki.

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
