## [Decision Letter · Decision Letter 0]

31 May 2024

Dear Dr Csankovszki,

Thank you very much for submitting your Research Article entitled 'Condensin IDC, H4K20me1, and perinuclear tethering maintain X chromosome repression in C. elegans' to PLOS Genetics.

The manuscript was fully evaluated at the editorial level and by independent peer reviewers. The reviewers appreciated the attention to an important problem, but raised some substantial concerns about the current manuscript. Specifically, the low sample size and biological replicates, as well as the lack of quantification of images, was a major concern, among other points raised by the reviewers. Based on the reviews, we will not be able to accept this version of the manuscript, but we would be willing to review a much-revised version. We cannot, of course, promise publication at that time.

Should you decide to revise the manuscript for further consideration here, your revisions should address the specific points made by each reviewer. Additionally, please provide correlation and clustering plots to evaluate quality of RNAseq data. We will also require a detailed list of your responses to the review comments and a description of the changes you have made in the manuscript.

If you decide to revise the manuscript for further consideration at PLOS Genetics, please aim to resubmit within the next 60 days, unless it will take extra time to address the concerns of the reviewers, in which case we would appreciate an expected resubmission date by email to plosgenetics@plos.org.

If present, accompanying reviewer attachments are included with this email; please notify the journal office if any appear to be missing. They will also be available for download from the link below. You can use this link to log into the system when you are ready to submit a revised version, having first consulted our Submission Checklist .

PLOS has incorporated Similarity Check , powered by iThenticate, into its journal-wide submission system in order to screen submitted content for originality before publication. Each PLOS journal undertakes screening on a proportion of submitted articles. You will be contacted if needed following the screening process.

We are sorry that we cannot be more positive about your manuscript at this stage. Please do not hesitate to contact us if you have any concerns or questions.

Yours sincerely,

Giovanni Bosco, Ph.D.

Section Editor

PLOS Genetics

Paula Cohen

Section Editor

PLOS Genetics

Reviewer's Responses to Questions

**Comments to the Authors:**

Reviewer #1: Trombley et al., characterize the role of condensing IDC at different stages of development in C. elegans. They show that while this complex, required for dosage compensation, is essential during embryogenesis, it is dispensable for survival in post-embryonic stages. To determine this, they make use of the auxin inducible degradation system, which allows to rapidly deplete a protein of interest, in their case DPY-27, upon exposure to the plant hormone auxin. They also investigate the role of a non-condensin DCC subunit termed DPY-21, which is a histone demethylase responsible for generating high levels of H4K20me1 on the repressed X chromosome in hermaphrodites, and of CEC-4, a H3K9me perinuclear anchor. The biological question addressed in this work is interesting and the genetic tools developed by the authors look convincing and suitable to their aims. However, the study at the current stage is very preliminary and requires substantial experimental work in order to become solid and convincing. Below, we outline the major points that in our opinion need to be addressed.

1. The approach of the authors relies mostly on immunostainings and X chromosome paints. Yet, all of the presented microscopy results come from a single biological replica where only about 20 nuclei per condition were analyzed. This is absolutely not sufficient to draw any meaningful conclusion. The authors should repeat their experiments to reach at least 3 biological replicas with the number of nuclei analyzed that should reach at least 50 in total. A possibility could be to quantify 15-20 nuclei per biological replica.

2. Besides the major issue raised in point 1, some of the microscopy data are not quantified at all. For example:

- In Fig. 2 A, B, C. no quantification of the degradation of DPY-27 is shown. Likewise, there is no quantification of CAPG-1 and H4K20me1. The authors write in the text that the signals of these factors become diffused. However, it is impossible to distinguish between a diffused signal vs loss of signal in the images presented. A quantification of the total fluorescent intensity/ per nucleus is needed to claim diffusion over degradation. Furthermore, it is possible that the signal decreases AND is diffused.

- In Fig. 4 A,B. Again there is no quantification of DPY-27 degradation. Also, there is no quantification of CAPG-1 and H4K20me1. The authors should do a colocalization analyses to prove that the effect on the distributionof these factors is partial after 1h on auxin.

- Fig.6 F, the authors state that loss of CEC-4 or DPY-21 “does not influence the localization of DPY-27 to the X”. However, there is no quantification for this statement. From the image presented, it looks like DPY-27 is more diffused in dpy-21 mutants. Thus, to draw meaningful conclusions, a colocalization analysis should be conducted.

3. In Fig.3, the authors perform a chromatin positioning assay quantifying the amount of X chromosome found in three concentric nuclear zones of equal surface. Such an assay does not directly address detachment of the X (more precisely the left tip of the X chr, see Ikegami et al., 2010) from the nuclear lamina as the current assay can be greatly influenced by X chromosome compaction, which is indeed affected in some of their conditions. For example, it is possible that the X chromosome, visualized with a whole chromosome paint, remains bound to the nuclear lamina but due to decompaction, reaches at least partially the internal zone of the nucleus. To be able to draw conclusions about the subnuclear distribution, or perinuclear positioning, of the X chromosome, the authors should use a different approach, such as LEM-2ChIP qPCR or LMN-1 or EMR-1 DamID amplifying sequences on the X, or, to stick to microscopy, a DNA FISH with a probe specifically targeting only the left tip of the X-chromosome, followed by the 3 zones assay or a measure of the distance from the nuclear envelope.

4. For all of the RNAseq data in Fig5 and 7, the authors should provide, as supplementary material, analyses showing the correlation and clustering between biological replicas.

5. For the analyses of Figs. 2, 3, 5 and 7 the authors compare worms that are likely at different developmental stages or worms with a different cell and nuclear size (as they show in Fig. 1D, that depletion of DPY-27 results in a stunted growth). While this might still be ok to allow drawing conclusions about the expression of X-linked genes vs autosomal genes (Figs 5 and 7), the ploidy and/or nuclear volume of animals exposed to auxin or not should be determined for the microscopy analyses. If the ploidy/nuclear volume ratio is the same between auxin and no auxin, then the samples can be compared. In contrast, different ploidy/nuclear volume ratios do not allow to conclude what determines the change in the amount of volume occupied by the X chromosome.

On this line, a proper positioning assay, as suggested in point 3, would instead not be affected by differences in ploidy/nuclear volume ratios as the authors would have a direct read out of the position of a smaller locus.

6. The role of the H3K9me reader CEC-4 in the dosage compensation process remains unclear. For example, in Fig. 4B, the authors show that H4K20me1 is still present on the X chromosome, yet the X chromosome is decompacted, as they previously showed, and in Fig. 3. A quantitative, ChIPseq (or qPCR targeting X-linked vs autosomal genes) approach is required to determine whether, despite being still present in IF, the levels of H4K20me1 are reduced or not in absence of CEC-4. To gain further mechanistic insights, the authors should also test the levels of H4K20me2/3 and H3K9me2/3 in presence and absence of cec-4 and dpy-21, with H3K9me2/3 being directly bound by CEC-4.

Minor points:

1. No auxin and auxin treatment should be in the same panel, as they need to be compared.

2. We find the choice of not using asterisks but A, B, C etc to indicate statistically significant differences rather confusing.

3. We suggest that the authors consider using boxplots rather than bar plots to better represent the variability of the data, particularly the microscopy ones.

Reviewer #2: In this study, Trombley et al. investigate the genetic requirements of X-chromosome dosage compensation during the maintenance phase in C. elegans animals. The authors focus on the roles of dosage compensation complex component DPY-27 and the chromodomain protein CEC-4 in intestinal cells. They find that loss of DPY-27 and CEC-4 in the maintenance phase lead to X-chromosome decompaction, H4K20me1 loss, and gene expression de-repression, but animals are still viable. The authors conclude that the DCC and lamina tethering consolidate X-chromosome repression in larvae. These findings are of interest to the chromatin biology community in general and dosage compensation field specifically.

To strengthen the conclusions of the manuscript and allow readers to assess the robustness/reproducibility of the data, I suggest the following:

- The experiments of Fig. 1 should have side-by-side quantification of the efficacy of DPY-27 depletion. DPY-27 stains should be performed on the same batches of embryos/larvae that phenotypes are quantified. How variable is the auxin depletion in the different conditions tested? Images from multiple replicates should be presented in the supplemental.

- All figure legends should state the number of biological and technical replicates performed.

- For all imaging data, uncropped and unaltered raw images should be provided in the supplemental along with their metadata. Some images in the figures appear to be over-processed, and the microscopy section of the methods is under-described. Exposure times and laser/LED power should be reported. All processing steps should be included – especially choice of nuclei to display, and all images should be displayed with identical min-max intensities per assay to have accurate comparisons. These values should be reported. Colorblind-friendly lookup tables should be used (e.g. do not combine red and green).

- For the gene expression analysis, it would be informative to dig a little deeper and un-collapse the gene expression plots to show how expression changes over the length of the X and autosomes. Do the regions overlap with H4K20me1? Known DCC ChIP regions? More details in the Methods section are required for ‘downstream analysis’ – which R packages? All scripts should be provided in supplemental/GitHub.

- I am curious about the difference between intestinal cells and other cells of the animal – are the conclusions held true? The extremely high ploidy of intestinal cells could be a factor in the degree of X-chromosome decompaction and H4K20me loss. Can the authors compare to muscle/vulva/neuron nuclei in their images?

- Clarify the discussion regarding the role of CEC-4 in embryos vs larvae, discussing previous literature on cec-4 loss (i.e. Gasser lab work, etc.). It is still unclear to me what the model for CEC-4 function is in different stages, and the difference between autosomes and X chromosome role. Perhaps add a final model figure.

**Have all data underlying the figures and results presented in the manuscript been provided?**

Reviewer #1: **No: ** As written in the comments to the author, they should provide correlation and clustering plots to evaluate the quality of the RNAseq

Reviewer #2: **No: ** Uncropped original images etc. for all figures should be provided. See specific comments in review.

PLOS authors have the option to publish the peer review history of their article (what does this mean? ). If published, this will include your full peer review and any attached files.

**Do you want your identity to be public for this peer review?** For information about this choice, including consent withdrawal, please see our Privacy Policy .

Reviewer #1: No

Reviewer #2: No

---

## [Decision Letter · Decision Letter 1]

17 Jan 2025

PGENETICS-D-24-00374R1

Condensin IDC, H4K20me1, and perinuclear tethering maintain X chromosome repression in C. elegans

PLOS Genetics

Dear Dr. Csankovszki,

Thank you for submitting your manuscript to PLOS Genetics. After careful consideration, we feel that it has merit but does not fully meet PLOS Genetics's publication criteria as it currently stands. Specifically, the use of the term "perinuclear tethering" seems inappropriate in that the data do not provide direct evidence of tethering per se as opposed to localization or positioning to the periphery as a consequence of compaction. Please change text in the title and body of the the manuscript so as to not indicate direct evidence for tethering, although you may suggest it in a hypothetical model. Additionally, their are numerous corrections that need to be addressed, as detailed by the reviewer comments. Because there are no new experiments or data required, I do not anticipate that a revised submission would have to go out again for review. Therefore, we invite you to submit a revised version of the manuscript that addresses the points raised during the review process.

Please submit your revised manuscript within 30 days Feb 16 2025 11:59PM. If you will need more time than this to complete your revisions, please reply to this message or contact the journal office at plosgenetics@plos.org. Please include the following items when submitting your revised manuscript:

We look forward to receiving your revised manuscript.

Kind regards,

Giovanni Bosco, Ph.D.

Section Editor

PLOS Genetics

Paula Cohen

Section Editor

PLOS Genetics

Aimée Dudley

Editor-in-Chief

PLOS Genetics

Anne Goriely

Editor-in-Chief

PLOS Genetics

**Reviewers' comments:**

Reviewer's Responses to Questions

**Comments to the Authors:**

Reviewer #1: I appreciate the efforts the authors have made to refine the manuscript, making it more precise and convincing. The revised boxplots provide much clearer insights into the image quantifications, and overall, the manuscript has improved. The conclusions that dosage compensation is not essential for life during post-embryonic stages and that CEC-4 exacerbates the phenotypes of DCC-impaired animals are well-supported.

However, I still have significant concerns regarding the conclusion that DPY-27 is required for subnuclear localization (title of subparagraph in lines390-391 and Figure 5).

On page 12, lines 452, and Figure 5C, I appreciate the authors discussing the role of decondensation. Nonetheless, I remain unconvinced that the FISH-based monitoring of the whole X chromosome provides sufficient evidence to claim subnuclear relocation over decompaction. In Figures 5A-B, the authors clearly show that the compaction of the X chromosome is affected by the loss of both CEC-4 and DPY-27. In Figure 5C, they quantify the percentage of the X chromosome found in three zones of equal surface area. Changes in these distributions do not necessarily reflect positioning alterations; rather, the increase in centrally located X chromosomes are likely a result of decompaction, which allows the chromosome to occupy more internal regions.

I remain convinced that, to substantiate their claims about perinuclear tethering (a term explicitly used in the title), the authors need to provide stronger evidence for their findings. As I have previously noted, they should focus on the left tip of the X chromosome, which is known to be anchored. While I understand the authors' response to my earlier comment that CEC-4 predominantly affects the middle part of the X and not the left tip (Snyder, 2016), this effect on the middle-X chromosome could still be explained by changes in compaction rather than positioning. What direct evidence is there, beyond the observation of a decompacted chromosome, that the X chromosome moves away from the nuclear periphery in any of their conditions? (less binding to lamin, LEM-2 or EMR-1?).

The authors mention in one of their responses to my previous comments that CEC-4 does not affect positioning in larvae, as other mechanisms are at play in post-embryonic stages. This is true, but then I wonder: why are they testing CEC-4 and claiming a change in perinuclear anchoring in larvae and adults? Based on Cabianca et al 2019, CEC-4 has also a role in anchoring chromosomes in differentiated cells, which is mostly evident on endogenous chromatin with EMR-1-DamID. Thus, I don´t argue against their choice of testing it. But while the observed differences are indeed intriguing, these can most likely be attributed to decompaction rather than true changes in subnuclear positioning, especially in absence of any other complementary experimental test of binding to the nuclear envelope.

Supplementary Figure 7 provides relevant data for Figure 5C, showing the distribution of the X chromosome across all three zones, which could help the authors make some conclusions on perinuclear positioning. First, to improve visualization, I suggest plotting the same genotypes with and without auxin side by side. For statistical analysis, I recommend conducting a chi-squared test to evaluate distribution changes across all zones, followed by a focused t-test to assess differences specifically in the outermost zone, similar to their approach for the central zone in the main figure. In fact, the critical question is: is there a statistically significant difference in the outer zone (perinuclear) when analyzed with a t-test?

If a significant difference is observed, I might agree that perinuclear anchoring is involved, as if chromosome X leaves the outermost zone, even when decompacted, this may indeed indicate a change in localizaiton, specifically a reduction in peripheral positioning. Without such evidence, the current data do not sufficiently support claims about perinuclear anchoring, subnuclear positioning, localization, or distribution and in that case, I would strongly recommend that the authors avoid describing their findings using these terms in their manuscript, unless additional evidence is provided. Even if the outermost zone occupancy is significantly different in DPY-27 perturbations, I suggest revisiting the title and removing the term "perinuclear tethering”, as some data would now be there, but not strong enough to appear in the title, as this could mislead the reader.

I appreciate the effort made to provide additional analyses and quality control data for their RNA-seq. However, their statement in Lines 476-479 is unclear. Which ones are the more limited number of samples? Which genotype is not robustly changing? Did they anyway include them in the analyses or exclude them?

Suppl. Fig13: please make this figure also in a version where, across panels, the data are plotted with the same scale bar. "This is fundamental for appreciating the relative contributions of the different perturbations in relation to one another.

Minor comments:

- For figures 2, 3, 4, the legend should indicate whether a single focal plane (of the many stacks they acquired) or a projection is shown. If single plane, how was it selected? In the middle? How do they guarantee picking unbiasedly? If projection, was it sum of stacks or maximum intensity or..?

- Fig1B: the titles of the graph have different size

- Line 273: a space is missing between In and TIR1

- Figure 3C and D: the asterisks are misaligned

- For fig 3D the description on the Y axis signal/background is unclear and lacking critical information. Is it mean fluorescence intensity? Maximum fluorescence intensity? Mean should be used.

- Line 389, please add “at least at the resolution scale of our Immuno Fluorescence assay”. Without other assays (e.g., ChIP) the authors cannot rule our effects of CEC-4 on H4K20me1.

- line 402, unclear what they mean with more diffuse with respect to the X in cec-4 mutants: less compact? If so, state that.

- Line 454, the term movement is not accurate as the authors do not perform any time lapse imaging. Please, revise.

- Line 784 “cec-4” should be italicized

Reviewer #2: Trombley et al. have made substantial revisions to their original manuscript to improve the robustness, transparency, and quantitative nature of their conclusions. My concerns have been largely satisfied (except one point below), and I find the choice of revisions by the authors reasonable.

My remaining concern is the interpretation of the CEC-4 results, which the authors can clarify with changes to the text. Since their previous results showed that the left arm of the X (where CEC-4 binds H3K9me) is not detached from the lamina in cec-4 mutants, and the current manuscript does not present results to contradict this, I believe the conclusions should be modified. As the other reviewer pointed out, loss of cec-4 could result in the decompaction of chromatin without loss of tethering. Moreover, the authors did not provide evidence that the H3K9me binding activity of cec-4 is required for the observed change in chromosome territory morphology. Some uncharacterized function of cec-4 independent of H3K9me binding on the left arm, or downstream of cec-4, may be aiding the compaction of the X globally. I do not think additional experiments to investigate this aspect are necessarily within the scope of this study. Therefore, the authors should avoid statements that link the phenotypes observed to K9me binding or lamina tethering and instead use more accurate descriptors like 'overall localization', 'territory volume/compaction', or similar.

Instances where corrections to cec-4 interpretation are needed:

- title 'perinuclear tethering'

- line 46

- line 170

- line 649

Additional point: Include the degree of X gene derepression in the abstract (line 42) or state 'partial derepression' to avoid misleading readers into thinking the Xs are completely derepressed.

**Have all data underlying the figures and results presented in the manuscript been provided?**

Reviewer #1: Yes

Reviewer #2: Yes

PLOS authors have the option to publish the peer review history of their article (what does this mean? ). If published, this will include your full peer review and any attached files.

**Do you want your identity to be public for this peer review?** For information about this choice, including consent withdrawal, please see our Privacy Policy .

Reviewer #1: No

Reviewer #2: No

**Figure resubmission:**
---

## [Editor Report · Decision Letter 2]

24 Feb 2025

PGENETICS-D-24-00374R2

Condensin IDC, DPY-21, and CEC-4 maintain X chromosome repression in C. elegans

PLOS Genetics

Dear Dr. Csankovszki,

Thank you for submitting your manuscript to PLOS Genetics. After careful consideration, we feel that it has merit but does not fully meet PLOS Genetics's publication criteria as it currently stands. Therefore, we invite you to submit a revised version of the manuscript that addresses the points raised during the review process.

As you will see, there are multiple reviewer concerns listed, although all these can be addressed by changes to your text (i.e. no new experiments are required). Specifically, the data do not support the claim of perinuclear tethering and therefore we ask that you change or soften your claims that localization or tethering to the perinuclear region is related to compaction. Please also see other specific points in the reviewer comments. I do not expect a revision to have to go out for review again if you are able to make these changes to your text.

Please submit your revised manuscript within 30 days Mar 25 2025 11:59PM. If you will need more time than this to complete your revisions, please reply to this message or contact the journal office at plosgenetics@plos.org. Please include the following items when submitting your revised manuscript:

We look forward to receiving your revised manuscript.

Kind regards,

Giovanni Bosco, Ph.D.

Section Editor

PLOS Genetics

Paula Cohen

Section Editor

PLOS Genetics

Aimée Dudley

Editor-in-Chief

PLOS Genetics

Anne Goriely

Editor-in-Chief

PLOS Genetics

**Journal Requirements:**

1) Please upload a copy of Figure S13C and D which you refer to in your text on page 14. Or, if the figure is no longer to be included as part of the submission please remove all reference to it within the text.

**Figure resubmission:**
---

## [Editor Report · Decision Letter 3]

28 Feb 2025

Dear Dr Csankovszki,

We are pleased to inform you that your manuscript entitled "Condensin IDC, DPY-21, and CEC-4 maintain X chromosome repression in C. elegans" has been editorially accepted for publication in PLOS Genetics. Congratulations!

Yours sincerely,

Giovanni Bosco, Ph.D.

Section Editor

PLOS Genetics

Paula Cohen

Section Editor

PLOS Genetics

Aimée Dudley

Editor-in-Chief

PLOS Genetics

Anne Goriely

Editor-in-Chief

PLOS Genetics

Comments from the reviewers (if applicable):

**Data Deposition**

http://datadryad.org/submit?journalID=pgenetics&manu=PGENETICS-D-24-00374R3

**Press Queries**

---

## [Editor Report · Acceptance letter]

PGENETICS-D-24-00374R3

Condensin IDC, DPY-21, and CEC-4 maintain X chromosome repression in C. elegans

Dear Dr Csankovszki,

We are pleased to inform you that your manuscript entitled "Condensin IDC, DPY-21, and CEC-4 maintain X chromosome repression in C. elegans" has been formally accepted for publication in PLOS Genetics! Your manuscript is now with our production department and you will be notified of the publication date in due course.

With kind regards,

Zsofia Freund

PLOS Genetics

On behalf of:
